# Functional analysis of the Aspergillus fumigatus kinome identifies a druggable DYRK kinase that regulates septal plugging

Norman van Rhijn [1,11], Can Zhao[1,11], Narjes Al-Furaiji [1,2,11], Isabelle S. R. Storer[1], Clara Valero[1], Sara Gago [1], Harry Chown [1], Clara Baldin [3], Rachael-Fortune Grant[1], Hajer Bin Shuraym [1,4], Lia Ivanova[5], Olaf Kniemeyer [5], Thomas Krüger[5], Elaine Bignell [1,6], Gustavo H. Goldman [7], Jorge Amich [1,8], Daniela Delneri [9], Paul Bowyer [1], Axel A. Brakhage[10], Hubertus Haas [3] & Michael J. Bromley [1] ✉

More than 10 million people suffer from lung diseases caused by the pathogenic fungus *Aspergillus fumigatus*. Azole antifungals represent first-line therapeutics for most of these infections but resistance is rising, therefore the identification of antifungal targets whose inhibition synergises with the azoles could improve therapeutic outcomes. Here, we generate a library of 111 genetically barcoded null mutants of *Aspergillus fumigatus* in genes encoding protein kinases, and show that loss of function of kinase YakA results in hypersensitivity to the azoles and reduced pathogenicity. YakA is an orthologue of *Candida albicans* Yak1, a TOR signalling pathway kinase involved in modulation of stress responsive transcriptional regulators. We show that YakA has been repurposed *in A. fumigatus* to regulate blocking of the septal pore upon exposure to stress. Loss of YakA function reduces the ability of *A. fumigatus* to penetrate solid media and to grow in mouse lung tissue. We also show that 1-ethoxycarbonyl-beta-carboline (1-ECBC), a compound previously shown to inhibit *C. albicans* Yak1, prevents stress-mediated septal spore blocking and synergises with the azoles to inhibit *A. fumigatus* growth.

Over three billion people are thought to be infected by filamentous fungi each year[1,2]. While the vast majority of these are relatively benign there are a significant number of more sinister infections that have high mortality rates and are exceptionally difficult to treat. By far the most prominent of these are caused by the genus *Aspergillus* and result in both chronic and invasive diseases that together account for more than 1.5 million deaths each year making them among the leading cause of infection driven mortality alongside tuberculosis and malaria[1]. The World Health Organisation has recently named *Aspergillus fumigatus*, the primary aetiological agent in aspergillosis, in the critical priority group of fungal pathogens[3].

Infections caused by *Aspergillus* typically originate in the lungs of susceptible individuals following inhalation of airborne fungal spores liberated from environmental sources such as compost heaps, soil and decaying vegetation. Despite the limited nutrient resources present in the lung, *A. fumigatus* spores are able to germinate and form hyphae that, in invasive disease, penetrate through the basal lamina, alveolar epithelium and basement membrane resulting in destruction of surrounding lung tissues and vasculature[4]. There are estimated to be over 300,000 cases of invasive infections every year with mortality rates ranging from 40 to 90% depending on disease setting[1,5,6]. With chronic forms, infections are typically restricted to cavities that have been

formed by previous damage, however over time these cavities can expand and limited invasion into surrounding tissue may occur[7]. Around three million people are thought to have a chronic form of the disease at any one time with around 5% succumbing to infection each year[1,7].

Our ability to treat these infections is hampered by a limited arsenal of effective antifungals[8,9]. The triazole class are currently used as first-line therapy for all forms of aspergillosis, however resistance develops frequently in chronic disease where treatments are given for a minimum of 6 months and are often extend well beyond this period[10,11]. The use of compounds analogous to the azoles for crop protection has also driven resistance in the environment which has resulted in significant increases in resistance in the clinic[12]. In some centres, resistance rates in invasive aspergillosis patients who have no prior antifungal treatment now exceed 20% and these patients are more likely succumb to infection even if alternate therapies are given[13]. The rapid emergence of azole resistance in the environment and during therapy has naturally driven the use of alternative therapies however these are either poorly tolerated, in the case of amphotericin B, or are perceived to be less effective and are not orally bioavailable[14]. Combination therapy as primary treatment may prove to be more effective in supressing resistance however this is untested and the aforementioned limitations of alternative therapies restrict treatment options[15,16]. Several novel antifungals are likely to be deployed clinically in the next few years however their development has not focussed on how and if they can be used in combination[8]. Notably we have recently shown that the activity of the novel clinical antifungal olorofim is antagonised by the azoles[17].

Here we report our efforts to identify genetic factors that facilitate adaptation of *A. fumigatus* to antifungal challenge and the mammalian host, specifically focussing on protein kinases, which are considered to be amenable for drug discovery purposes[18]. We have generated a collection of 111 genetically barcoded protein kinase null mutants in *A. fumigatus* and, using Bar-seq mediated competitive fitness profiling reveal a critical role for the dual-specificity tyrosine-regulated kinase (DYRK) YakA in adaptation to iron limitation, temperature, pH, hypoxia and azole drug stress and in contrast to the role played by its orthologue in yeast-type fungi, pathogenicity in a mouse model of infection. Our data shows that, distinct from its function in yeasts, YakA has a non-redundant role in regulating adaptation to these environments by phosphorylation of proteins that control the plugging of fungal septa or cross-walls that allow compartmentalisation of hyphae. In addition to the inability to adequately adapt to micronutrient and drug stress, the lack of YakA mediated compartmentalisation of hyphal filaments prevents *A. fumigatus* from building sufficient turgor pressure to facilitate penetration into solid substrates. Finally, we show how a beta-carboline DYRK inhibitor can selectively inhibit the action of YakA to prevent septal plugging, penetration and potentiate the action of other antifungals in *A. fumigatus*.

## Results

### Annotation of the protein kinases of *A. fumigatus* and generation of a barcoded null mutant library

The protein kinase (PK) complement in *A. fumigatus* has been defined either from homology to annotated proteins in *Aspergillus nidulans*[19] or bespoke assessment of now superseded annotations of the *A. fumigatus* genome using Hidden Markoff Models (HMM)[20]. We therefore performed a de novo evaluation of a recently updated annotation of the *A. fumigatus* (Af293) genome using the HMMer and the kinomer HMM Library[21] to identify protein kinases. The hits from this library were supplemented with PKs identified via semantic searches of the annotated genome databases at AspGD and FungiDB. A total of 133 unique PKs were identified using kinomer with an additional 19 (all of which were histidine kinases) being found through database searches

(Supplementary Data 1). An assessment of the genome of another sequenced strain of *A. fumigatus*, A1163 revealed the presence of 163 protein kinases. Pairwise comparison of the two PK cohorts revealed that 2 of the protein kinases identified in Af293, did not have an annotated orthologue in A1163 (Afu7g04735, Afu1g16000). BLASTn analysis of the A1163 genome with the Afu1g16000 and Afu7g04735 coding sequence revealed regions sharing 95% and 100% sequence identity respectively indicating missing annotations from this genome. Interestingly, comparison of mapped RNAseq data to the A1163 genome indicates that Afu7g04735 has been misannotated in Af293[22]. When corrected, Afu7g04735 and Afu1g16000 share (98%) sequence identity and may be the result of a recent gene duplication event. Ten protein kinases from A1163 did not have reciprocal BLAST hits in Af293. Of these, 2 (AFUB_079830 and AFUB_071620) seemed to be duplications of another protein kinase (AFUB_044560; Afu3g03740), 3 (AFUB_044400, AFUB_077790 and AFUB_044260) had regions of the genome that share >90% sequence identity with Af293 however the ORFs appear to have been truncated while 5 (AFUB_071600, AFUB_045710, AFUB_044400, AFUB_075350 and AFUB_081220) were missing altogether from the Af293 genome.

The disparity in the conservation of the PK cohort between the 2 strains led us to assess if these proteins were conserved across 218 *A. fumigatus* sequenced isolates[12]. Of 155 protein kinases in A1163 we assessed further, 135 were considered core (i.e. found in >95% of all genomes) while 20 kinases were considered accessory genes. In an attempt to assess the evolutionary lineage of the non-mammalian kinases, a phylogenetic tree was built (Fig. 1A). Consistent with our classification data from Kinomer, the vast majority of the kinases that fall into specific kinase categories clustered together. Two clear evolutionarily distinct clusters were identified, encapsulating kinases Afu3g13210, Afu3g08710, Afu6g03252, Afu5g15080 (Cluster FF1) and Afu1g16060, Afu5g14970, Afu3g02740 Afu6g03240. All of these protein kinases are conserved in A1163.

To construct a null mutant library of the PKs we had identified, we chose to employ a fusion PCR approach described by Zhao et al. and previously used to generate a 484-member library of transcription factor null mutants[22,23]. A unique 20-bp barcode sequence flanked by universal regions was introduced at the time of strain disruption to enable assessment of fitness in pooled assays using bar-seq. Cassettes were successfully amplified for 155 kinase genes, and were used to transform MFIG001, a Δ*ku80*, *pyrG*+ strain derived from FGSC strain A1160[24]. We isolated homokaryotic null mutants for 111 (67.5%) protein kinase genes (Fig. 1A) but despite several attempts (minimum *n* = 3), we were unable to isolate homokaryotic null mutants for 44 kinases however we were able to identify balanced heterokaryons for 40 (Supplementary Data 1). Of the protein kinase genes that we were unable to generate a homokaryotic null mutant for, all were considered core kinases with 30 previously being identified as essential or conditionally essential in *A. nidulans* or *A. fumigatus*[25,26]. Our inability to isolate homokaryotic null mutants is indicative of gene essentiality. To verify this, strains carrying tetracycline repressible alleles were generated for 4 of the genes in our list, all of which lacked growth in the presence of doxycycline (Fig. 1B). For 18 protein kinases, gene disruption (rather than replacement) mutants have previously been obtained by using a CRISPR-Cas9 mediated knock-in approach[27] indicating that the strains generated in that study may have retained partial functionality.

### High throughput competitive fitness reveals protein kinases required for *A. fumigatus* adaptation to stress and virulence

The introduction of unique genetic barcodes in our *A. fumigatus* null mutant collection enabled us to perform competition experiments between strains. Using procedures that limit hyphal fusion and nuclear exchange between strains[28] we carried out in vitro competitions in 12 different growth conditions that induce micronutrient (iron),

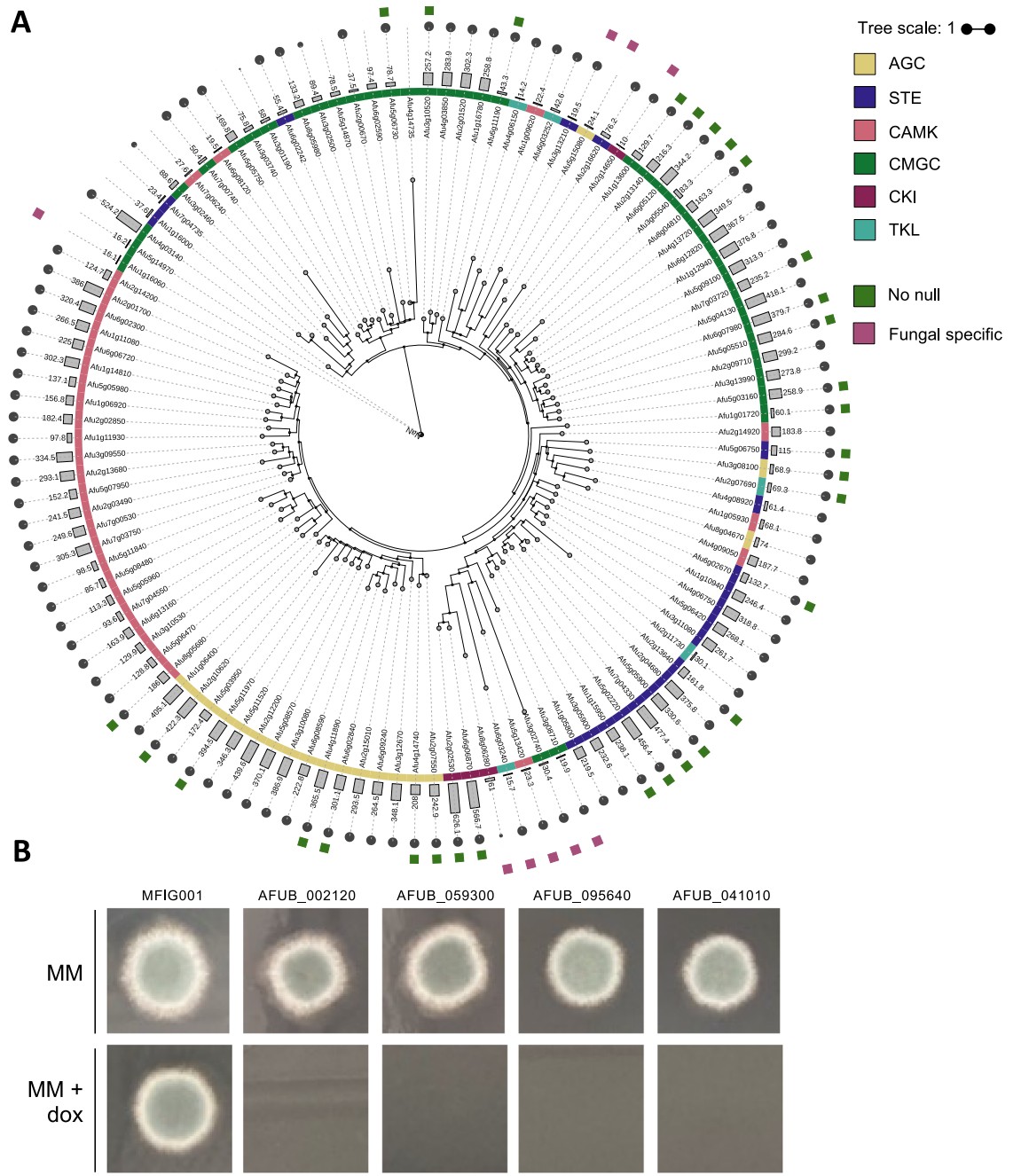

**Fig. 1 | Generation of the kinase knockout collection in *A. fumigatus*. A** A phylogenetic tree (Maximum-likelihood) of the protein kinases identified in Af293. The protein kinase family as determined using Kinomer v1.0 (first colour ring), the Kinomer score (bar chart with higher values representing higher confidence matches), the occurrence of each kinase within the pangenome of *A. fumigatus* (circle size with larger circles representing kinases that occur more frequently), if a homokaryotic null was constructed (green squares) and if the protein kinase is fungal specific (purple square) is shown as annotations around the phylogenetic tree. **B** Four putative essential kinases were validated using a Tet-OFF construct to replace the native promoter. Shutting down expression via addition of doxycycline to the medium resulted in no growth after 72 h at 37 °C, adding further evidence to the essentiality of these kinase for viability.

temperature, pH, antifungal and oxidative stress (Fig. 2A). Hierarchical clustering of this quantitative pheno-profiling data revealed functionally supported links between MAP kinases with known regulatory connectivity. Specifically, the cell wall integrity signalling cascade, which is comprised of *mpkA* (AFUB_070630), *mkk2* (AFUB_006190) herein named *mkkA* to conform to *A. fumigatus* gene nomenclature standards and *bck1* (*bckA*; AFUB_038060), form a compact cluster by virtue of their reduced ability to adapt to changes in pH and temperature[29]. Interestingly, the phenotypes exhibited by the *bckA* null mutant were significantly less pronounced than that of the downstream targets of the cascade when placed under iron stress. This

is consistent with previous reports that the phenotype of a *bckA* disruptant is less severe[27], and indicates a level of redundancy or functional divergence at the start of this signalling module. The two other well defined MAP kinase signalling cascades (High osmolarity and pheromone response pathways) failed to cluster either because members were missing from the null mutant collection (*mpkB* and *pbsB*) or we did not screen under conditions likely to exhibit a strong phenotype. Interestingly functional clusters were identified including *halA* (AFUB_053500), an orthologue of the *S. cerevisiae* salt tolerance kinase HAL4 and *skyA* (AFUB_096590). Both have growth defects in the high salt containing Aspergillus Minimal Media (AMM) and a

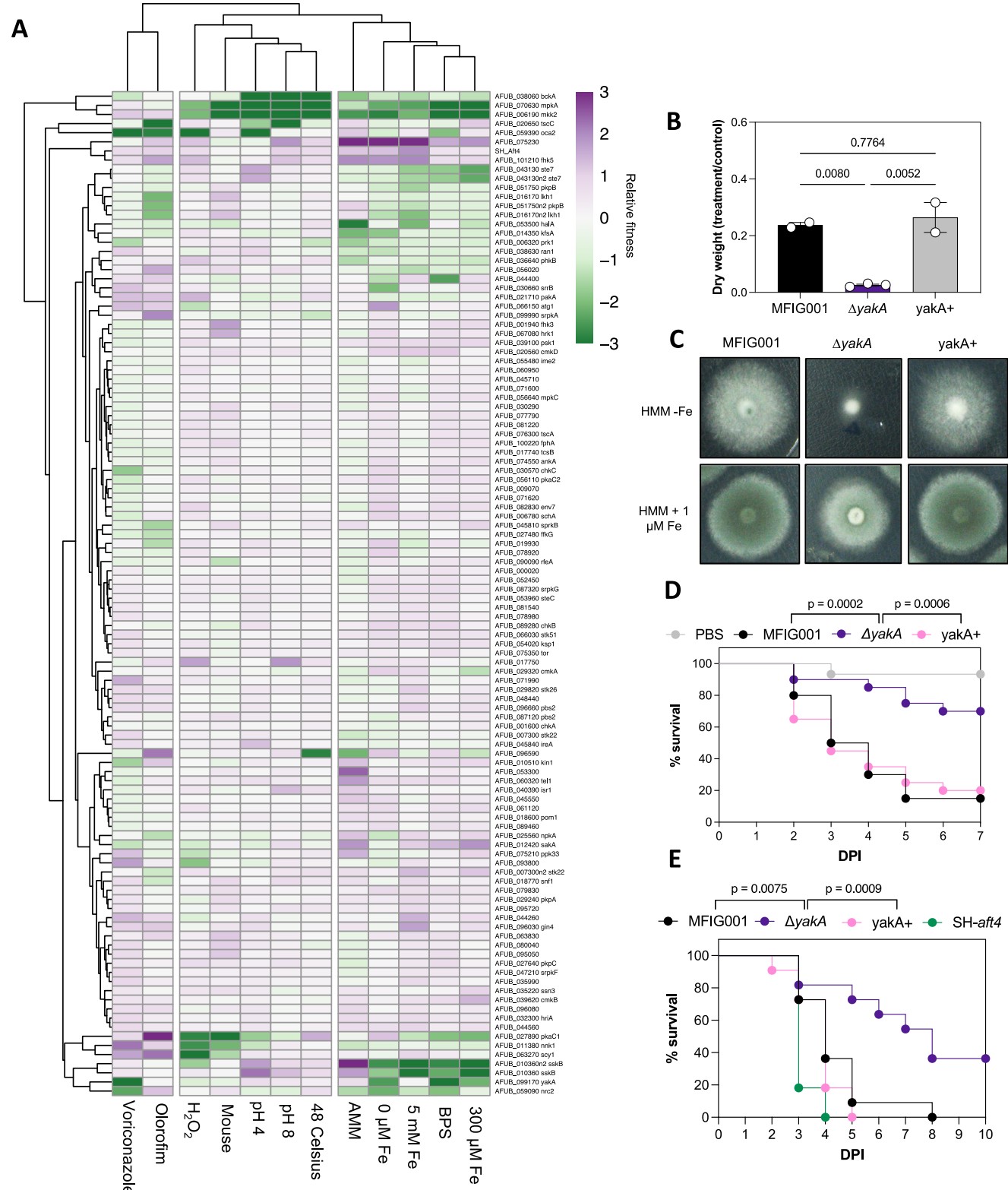

competitive advantage in low iron environments however the Δ*skyA* alone has a distinct defect in adaptation to high temperature.

The barcoded collection also afforded the opportunity to quantitatively assess competitive fitness of isolates within a mouse model of infection. Of the strains assessed, 3 protein kinase mutants had a demonstrable and statistically supported reduced fungal burden in mice with chemically induced leukopenia, namely Δ*pkaC* (AFUB_027890;) which had a mean log2 fold change in normalised

barcode counts of −9.7, Δ*mpkA* (−14.3 Log2FC) and Δ*mkkA* (−12.8 Log2FC)[30]. A further 9, Δ*bckA* (ΔAFUB_038060), Δ*rfeA* (AFUB_090090), Δ*scy1* (ΔAFUB_063270), ΔAFUB_075230, Δ*chkb* (ΔAFUB_089280), Δ*nnk1* (ΔAFUB_011380), Δ*halA* (ΔAFUB_053500), ΔAFUB_044400, and ΔAFUB_099170 (an orthologue of the *yak1* protein kinase of *S. cerevisiae* and herein defined as *yakA*) had notable (<−1.4 Log2FC) reductions of relative fungal burden however they failed to reach statistical significance in this experiment

**Fig. 2 | Bar-seq of *A. fumigatus* identifies YakA as a potential antifungal target.**
**A** Heatmap of fitness for each protein kinase null mutant relative to control conditions ($n = 5$). Relative fitness values was calculated by DESeq2, comparing barcode counts obtained from 20 h fRPMI1640 (pH 7, 37 °C, shaking 130 rpm) cultures with those supplemented with 1.5 mM $H_2O_2$ (to induce oxidative stress), voriconazole 0.15 mg/l, olorofim 0.002 mg/l, pH adjusted to pH4 or pH8 or with modified culture temperature (48 and 30 °C). and different iron concentrations (Fe−, BPS, 30 μM, 300 μM, 5 mM). The mouse data fitness values was calculated by DESeq2, comparing barcode counts obtained from the input pool with the infected samples. The heatmap was generated with Pheatmap, full linkage clustering on rows and columns (cutree_cols = 2). Source data are provided as a Source Data file. **B** Dry weight was measured after 16 h culture in HMM, with and without 0.5 mg/l itraconazole, at 37 °C shaking 130 rpm. Data are presented as mean values with SEM. Statistical difference was assessed by using one-way ANOVA ($n = 2$ or 3 biological replicates). Source data are provided as a Source Data file. **C** 1000 spores of MFIG001, Δ*yakA* and the reconstituted isolate (yakA+) were spot inoculated on HMM agar without iron and with 1 μM iron added and incubated for 72 h at 37 °C. Representative pictures are shown. **D** Survival curves in a *Galleria mellonella* model of aspergillosis. Statistical difference was assessed by Kaplan−Meier survival analysis and *p* values between groups are shown on the graph. **E** Survival in a leukopenic mouse model of aspergillosis ($n = 11$). Statistical difference was assessed by Kaplan−Meier survival analysis *p* values between groups are shown on the graph.

(Supplementary Data 2 and Supplementary Fig. 1). Crucially however we were able replicate and statistically verify the modest defect in virulence we observed in the mouse competition infection model for the *yakA* null mutant in a *Galleria mellonella* model of aspergillosis and a leukopenic mouse model of aspergillosis when they were challenged with single isolates (Fig. 2D, E). The indicative loss of fitness of the *yakA* mutant in the mouse lung, an environment known to be micronutrient limited, correlated with a significant reduction in fitness in low iron environments (−1.15 log2FC in 0 mM Fe AMM, −2.96 log2FC when BPS was added as chelator) but not under control growth conditions (RPMI-1640; 37 Celsius pH 7.0). The *yakA* mutant was of further interest as it also exhibited a striking reduction in competitive fitness when placed under azole stress (−4.6 log2FC; *p*adj $3.99 \times 10^{-34}$) marking it as a potential target for the development of an antifungal that could both treat infection and work synergistically with the azole class of antifungals that represent the current gold standard treatments for all forms of aspergillosis.

Comparative phenotyping was performed to assess the extent of the defects observed in the Δ*yakA* mutant. Biomass was reduced by 89% when compared to the wild-type isolate upon exposure to 0.5 mg/l of itraconazole while the MIC was reduced by 2-fold (Fig. 2B). In solid media, growth of the *yakA* mutant was significantly impaired in low iron and zinc environments and was notably reduced under high pH stress, consistent with the reduced solubility of both iron and zinc in alkaline environments (Fig. 2C). Interestingly, and contrary to our expectations, Δ*yakA* did not show any discernible growth defect under iron limitation in static liquid medium (Supplementary Fig. 2A), nor did it differ in in siderophore production (Supplementary Fig. 2B). These findings led us to characterise the function of YakA in more detail.

## YakA is required to facilitate penetration and establish turgor pressure

YakA is a DYRK kinase, which has been described in several other fungi including *C. albicans*, *S. cerevisiae*, *Fusarium* species, *Penicillium* species, *Aspergillus nidulans*, *Magnaporthe oryzae* and *Botrytis cinerea*[31–35] and been shown to function by phosphorylating transcription factors in response to stress. Specifically in *S. cerevisiae*, the orthologue *yak1* is under control of the Tor kinase and a 14-3-3 protein Bmh1 and is imported into the nucleus in response to various stressors which initiates the activation of the transcriptional regulators Haa1, Pop3, Hsf1, Msn2[36,37] Nuclear import is governed by two nuclear location signals (NLS) and adjacent phosphorylation sites. Notably it has been suggested that in *Fusarium graminearum* YakA phosphorylates HapX, a transcription factor responsible for regulating iron homoeostasis in filamentous fungi[38].

Alignment of YakA with the *S. cerevisiae* orthologue reveals that one of two proposed NLSs (NLS1 [123]RRRK[129]) and its downstream phosphorylation sites S[127+128] are lacking in YakA and only partial conservation of NLS2 ([289]PKFRR[293]), but not its adjacent phosphorylation sites, is evident[36] (Supplementary Fig. 3). Furthermore, while there is significant conservation of the protein kinase domain (58%) conservation at the region associated with Bmh1 regulation (31%) and the

C-terminal (7% over 206 AA) and N-terminal regions (21% over 200 AA) is very limited. This led us to hypothesise that the *A. fumigatus* protein domain structure differs from *S. cerevisiae* and might therefore have different or additional functions.

Consistent with the role of Yak1 in sensing carbon starvation, Δ*yakA* was unable to effectively utilise xylose and fructose and had severe growth defects in the non-fermentable carbon source glycerol, acetate and ethanol (Supplementary Fig. 4). Furthermore, and inconsistent with a role for YakA in generalised activation of HapX, Δ*yakA* had no defect in iron chelating siderophore production in low iron environments (Supplementary Fig. 2B).

Interestingly, the stress induced growth defects we observed in the *yakA* mutant was more profound when we increased the agarose concentration of solid medium (Fig. 3A). Three-dimensional tracking of fungal hyphae growing into solid medium showed that Δ*yakA* was able to penetrate through the media at 3% agarose, but not at 4% (Supplementary Fig. 5A). This phenotype was exacerbated under iron limitation. This led us to investigate if the inability of the Δ*yakA* mutant to penetrate the mammalian lung during infection was linked to the reduction in virulence. Cross sections of infected lung tissue revealed large lesions in wild-type and the genetically reconstituted mutant strain with hyphae appearing throughout the lung (Fig. 3B). However, for the *yakA* mutant, only small lesions were evident in infected lungs, and consistent with our hypothesis, limited penetration of the squamous epithelial layer into the surrounding lung tissue was observed (Fig. 3C). Interestingly we identified limited instances of airway plugging caused by the *yakA* mutant growing within alveoli (Supplementary Fig. 5B) indicating a likely cause of mortality in mice infected with the *yakA* mutant.

To penetrate stiff substrates filamentous fungi require the build-up of significant turgor pressure within the hyphae[39]. We therefore assessed the ability of Δ*yakA* to re-establish turgor after cytorrhysis[40]. After induction of cell collapse, the wild-type isolate was able to fully recover within 8 min however, the Δ*yakA* membrane collapsed further and was unable to fully recover, even after 20 min (Fig. 3C) suggesting loss of penetrative ability could be linked to failure to establish sufficient turgor pressure.

## YakA regulates phosphorylation of proteins associated with the septal pore

There was a clear disconnect between the inability of the *yakA* null to grow effectively in low iron environments, its perceived role in the regulation of the iron homoeostasis transcription factor HapX and its unaltered ability to regulate the production of siderophores. In an attempt to uncover why this could be we assessed phosphoregulatory role of *yakA* under iron limiting conditions (Fig. 4A). Overall, 18,076 phosphopeptides were detected from 2929 proteins (Supplementary Data 3). K-means clustering ($n = 10$) of differentially phosphorylated peptides under iron limitation revealed clusters of peptides that were regulated in an YakA dependent ($n = 1174$ downregulated−cluster 1, $n = 330$ upregulated−cluster 2 (Fig. 4A)) and YakA independent manner ($n = 557$ downregulated−cluster 3, $n = 434$ upregulated−cluster 4)

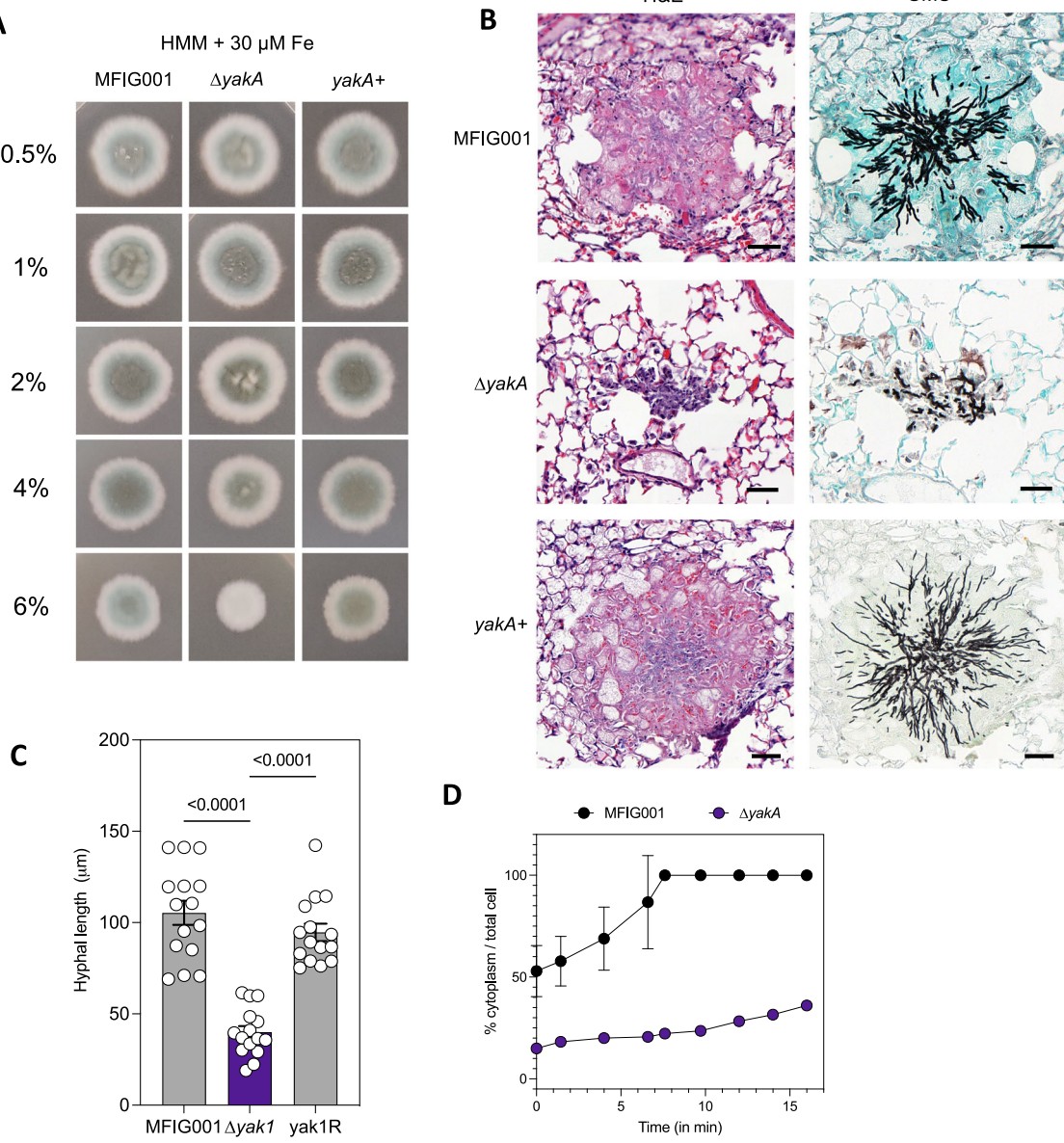

**Fig. 3 | Detailed phenotypic analysis of the ΔyakA isolate. A** $10^3$ spores of the isotype control MFIG001, ΔyakA, yakA+ were spotted on HMM + 30 µM Fe with increasing concentration of agarose (0.5–6%) and imaged after 72 h incubation at 37 °C. Representative images are shown. **B** Representative histological sections of lesions found in mouse lungs infected for 3 days with MFIG001, ΔyakA and yakA+. H&E and GMS stains were performed on neighbouring sections. **C** The hyphal length within histological sections ($n$ = 15 biological replicates) was measured. Data are presented as mean values with SEM. Statistical differences were assessed by one-way ANOVA. Source data are provided as a Source Data file. **D** Recovery from cytorrhysis after addition of glycerol to liquid cultures. Liquid cultures were grown for 16 h at 37 °C in an Ibidi imaging chamber. 1.8 M glycerol was added and images were taken every 150 s. The % cytoplasm compared to the total cell size was measured for individual septal compartments ($n$ = 10 biological replicates) using Fiji. Data are presented as mean values with SEM. Source data are provided as a Source Data file.

(Supplementary Data 3 and Fig. 4A). Consistent with our observations that siderophore levels were unchanged in the YakA null mutant, peptides derived from siderophore and iron transporters MirB, Sit1 and Sit2 had higher levels of phosphorylation under iron deplete conditions (MirB: 399 log2 fold up, Sit1: 8.3 log2 fold up, Sit2: 4.1 log2 fold up, MirC: 5.0 log2 fold up) but were not affected by loss of YakA (cluster 4; Fig. 4A). Peptides from proteins involved in siderophore biosynthesis were also found within cluster 4 (SidT: 692 fold up, SidA: 4.8 fold up, SidD: 240 fold up, SidG 6.1 fold up, SidL: 4.55 fold up,) (Supplementary Data 2).

A GO-term analysis (cellular component) of iron regulated phosphorylation that was dependent on YakA included proteins involved in cell morphology, division, the septum, membrane and cellular homoeostasis (cytoskeleton, vesicle and nuclear pore) (Fig. 4B). An association between YakA and phosphorylation of proteins associated with the septum, a cross-wall structure that divides hyphal cells and is required for the generation of turgor pressure[41,42], could explain the penetration defect observed in the ΔyakA strain. A more detailed analysis of proteins known to be involved in septal formation showed that those with the biggest phosphorylation changes are essential for septal plugging (Fig. 4C). Notably, these include the leashin protein Lah which tethers a plug known as the Woronin body to the septa, a septal pore associated protein, Spa10 which stabilises Lah at the septal pore and Soft (SO) a scaffolding protein associated with the septal pore upon cell wounding[41]. These results suggested that YakA has a critical role in regulating septal plugging under stress.

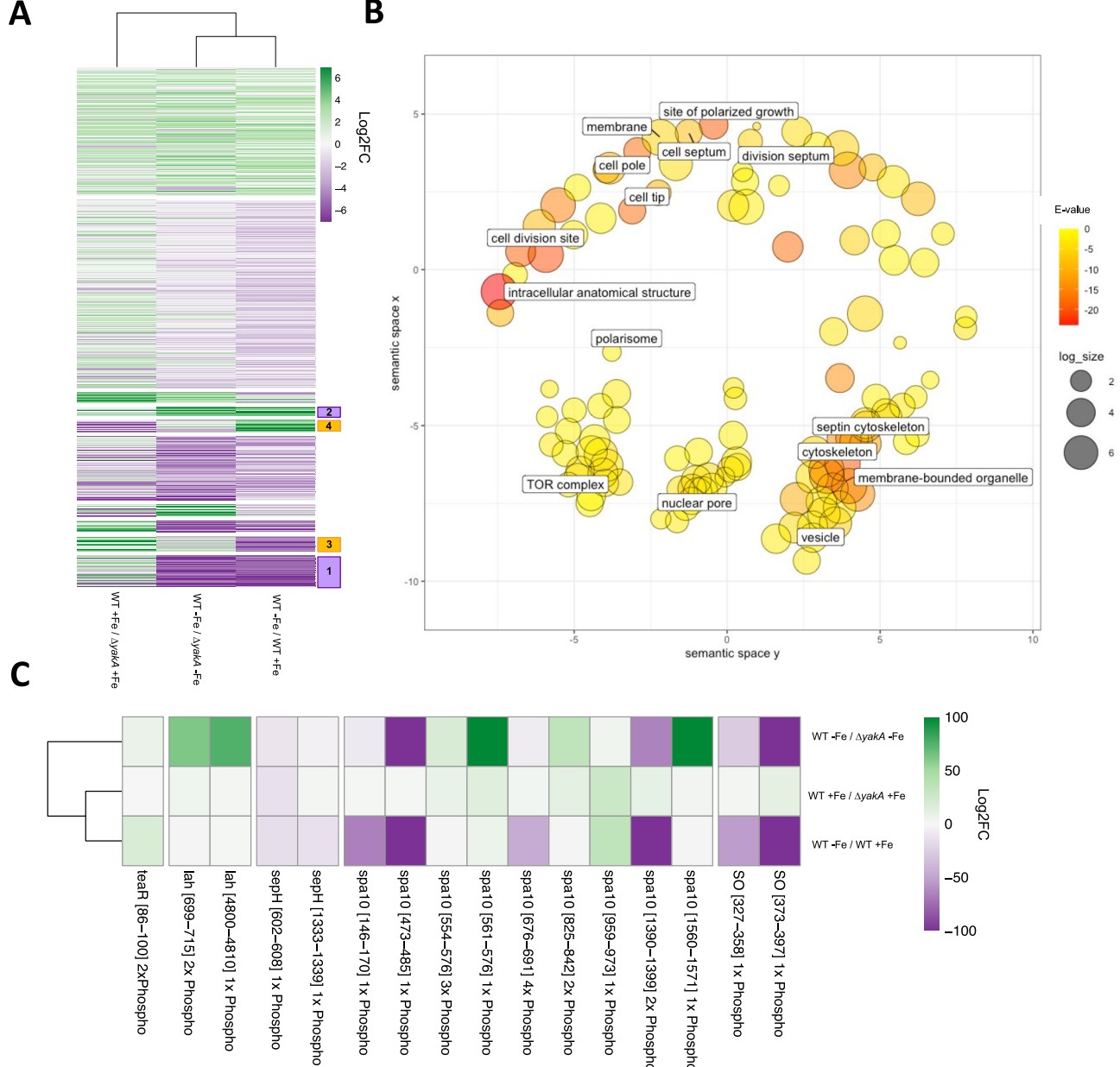

**Fig. 4 | Phosphoproteomics in response to iron limitation. A** A heatmap showing associations between changes in phosphopeptide abundance between MFIG001 and Δ*yakA* or upon low iron stress. K-means clustering was performed to identify clusters of phosphopeptides differentially phosphorylated (1 + 2 iron responsive and YakA dependent; 3 + 4 iron responsive YakA independent). **B** GO-term analysis in Revigo of *yakA*-dependent phosphopeptides (clusters 1 + 2) in response to iron limitation. Size of circles is the size of each enrichment category. Colour represents the *p* value. **C** An analysis of phosphopeptides that are present as components of the septal pore. Only phosphopeptides that are differentially regulated in at least one condition are shown (>2-fold up or down).

## YakA localises to the septal pore and regulates septal plugging by HexA and Lah

In order to assess the cellular localisation of YakA we generated a strain expressing a YakA-GFP fusion strain. YakA-GFP signal was detectable but dim under iron replete medium (Fig. 5A). We hypothesised that similarly to Yak1 in *S. cerevisiae*, we would observe a response when adding rapamycin, which inhibits the action of TOR, a known negative regulator of Yak1. In line with previous findings, YakA-GFP signal became brighter upon rapamycin addition however unlike in *S. cerevisiae*, YakA did not localise to the nucleus but consistent with our phosphoproteomic data localised to the centre of fungal septa at the presumptive septal pore (Fig. 5A, B). Furthermore, the increase in apparent abundance of YakA and septal localisation was induced by iron limitation (Fig. 5A) and YakA was required for susceptibility to the

echinocandin and azole class of antifungals (Supplementary Fig. 6A–C).

Our data led us to explore the role of YakA in the localisation of the Woronin body associated protein HexA and its tether Lah. HexA, which encodes a C-terminal peroxisomal targeting sequence (PTS1), localises within peroxisomes during its inactivated state, but localises to the septal pore when hyphae are damaged[35], preventing loss of cytosol from neighbouring compartments. Lah localises to the septal pore and associates with Woronin bodies, with its C-terminal governing the former and its N-terminal the latter (Supplementary Fig. 7). In keeping with previously published data[37], a C-terminally GFP tagged HexA (HexA:GFP) which lacks a functional PTS, localises across septa while in its inactivated state. Here we show that upon iron limitation, HexA:GFP localises to the septal pore (Supplementary Fig. 7) as does

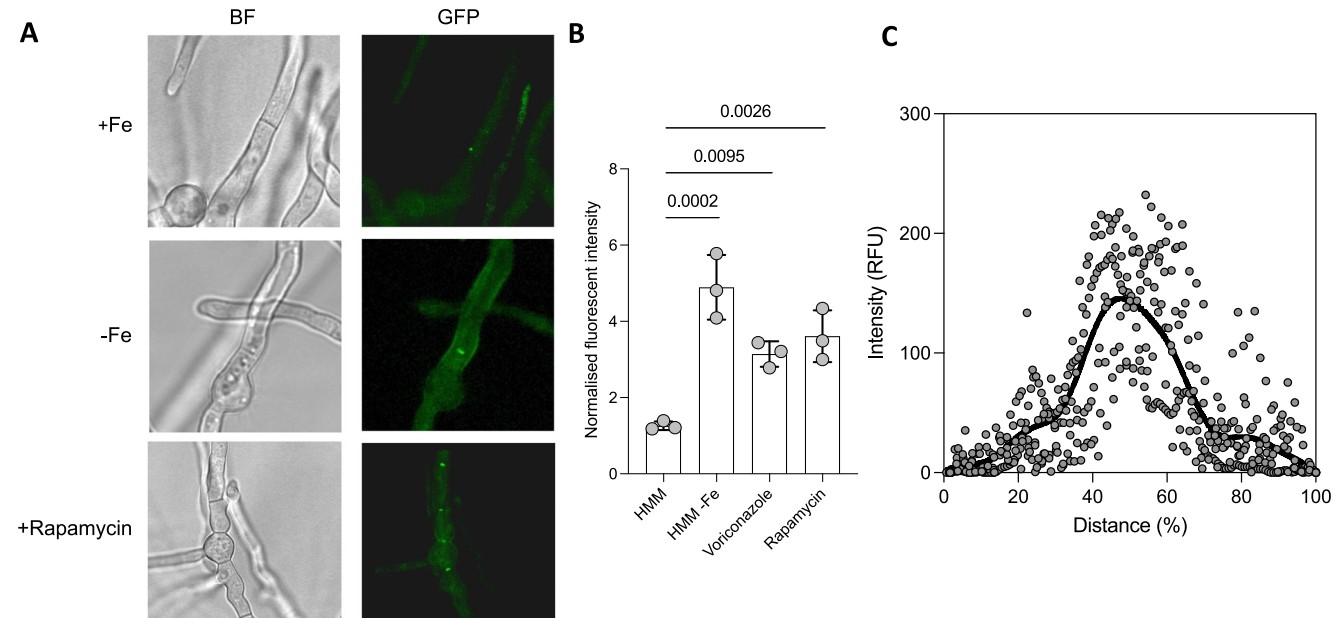

**Fig. 5 | YakA localisation to the septal pore. A** YakA-GFP localisation following overnight growth (16 h) in AMM, and replacement with either AMM (+Fe), iron limiting AMM (−Fe), or AMM supplemented with rapamycin for 1 h. Localisation of the GFP signal can be observed at the centre of septa. Scale bar = 10 μm. **B** The total fluorescent intensity was measured normalised to the background intensity following overnight growth (16 h) in AMM, and replacement with either AMM (+Fe), iron limiting AMM (−Fe), or AMM supplemented with rapamycin for 1 h. Quantification was performed using Fiji (n = 3 biological replicates). Source data are provided as a Source Data file. **C** Quantification of signal through a cross-section of septum shows higher fluorescent intensity in the central region of the septum, indicating septal pore localisation of YakA-GFP (n = 7). Source data are provided as a Source Data file.

the C-terminal domain of Lah (LahC:GFP) (Fig. 6A). Deletion of *yakA* in the LahC:GFP background resulted in complete loss of the low iron induced GFP signal from the fungal septum. Deletion of *yakA* in HexA:GFP[25] results in the GFP signal became less centrally located at the septal pore and more widespread across the fungal septum (Fig. 6B), phenocopying what is seen in this genetic background when the C-terminal domain of Lah is disrupted. Together these data suggest that YakA activity is required for micronutrient stress-initiated localisation of Lah and HexA at the septal pore and subsequent septal plugging.

### YakA can be inhibited by the action of beta-carbolines to prevent septal plugging and potentiate azole action

We hypothesised that YakA could function as an antifungal target as it is required for pathogenicity and susceptibility to the azole and echinocandin class of antifungals (Supplementary Fig. 6). We therefore explored the druggability of YakA by performing a virtual screening of small molecules against an AlphaFold predicted structure of YakA (Fig. 7A). Through blind docking, clusters of ligands binding to three pockets were identified, one in the active site and two smaller clusters within the C-lobe of the protein[43]. Of these three sites, the most probable druggable pocket was within the active site (0.81 druggability score as defined by PockDrug). This pocket consists of 30 amino acid residues with a volumetric hull of 2363.73A. In a previous study, the Yak1 protein of *C. albicans* was shown to be inhibited by the beta-carbolines 1-ECBC and 1-ABC, the latter being a small molecule produced by *Lactobacillus* species. 1-ECBC bound with high affinity (ΔG: −7.7 kcal/mol; Fig. 7A) to the active site pocket of the predicted *A. fumigatus* YakA protein[44].

To obtain evidence that YakA can be chemically inhibited, we challenged *A. fumigatus* with 1-ECBC under iron replete and limiting conditions and assessed hyphal length and the impact of localisation of YakA and Lah (Supplementary Fig. 8A and Fig. 7B). Upon iron limitation 1-ECBC blocks YakA recruitment to the septal pore (Fig. 7B and Supplementary Movie 1 and 2). Interestingly, this phenomenon was

only observed when YakA was not already present at the septal pore indicating that septal pores cannot be unblocked by disrupting YakA function (Supplementary Fig. 8B). As the Δ*yakA* mutant is hypersensitive to the azoles we explored the combinatorial effect of voriconazole and 1-ECBC to inhibit growth of MFIG001 (Fig. 7C). A strong synergistic effect (FICI = 0.28) was observed in a checkerboard assay.

Taken together, we have shown that generation of mutant collections for the purpose of competitive fitness profiling can be deployed to identify antifungal drug targets in *A. fumigatus* using both in vitro and in vivo assays. Using this strategy, we have identified YakA, a DYRK kinase, that is required for adaptation to azoles, iron homeostasis through its role as a mediator of septal plugging. We also demonstrate YakA is a druggable kinase, which can be targeted by natural products of the beta-carboline class to prevent septal plugging and potentiate the action of the azoles in *A. fumigatus*.

## Discussion

A dearth of antifungal agents and pandemic of antifungal drug resistance is hampering treatments of aspergillosis[9,45]. It is critical that strategies are developed to improve treatment outcomes and development of novel antifungal agents is a key way this objective will be achieved[9]. Here, through our assessment of the kinome of *A. fumigatus* we have shown how high throughput functional genomics can enable the detection of novel targets that co-ordinately regulate adaptation to the host environment and antifungal challenge.

Competitive fitness profiling of large knockout libraries has been used to great effect in microbes to identify hitherto unknown critical functions for genes in adaptation to numerous stressors[46,47]. More recently these approaches have taken advantage of massively parallel sequencing technologies to assess fitness of barcoded collections in yeasts[26]. There are however difficulties with applying competitive fitness profiling to filamentous fungi as hyphae of isotype strains can anastomose (fuse), allowing exchange of genetic material between hyphae and the mixing of nuclei between cells[28]. This is compounded by the fact that unlike yeasts, filamentous fungi compromise of

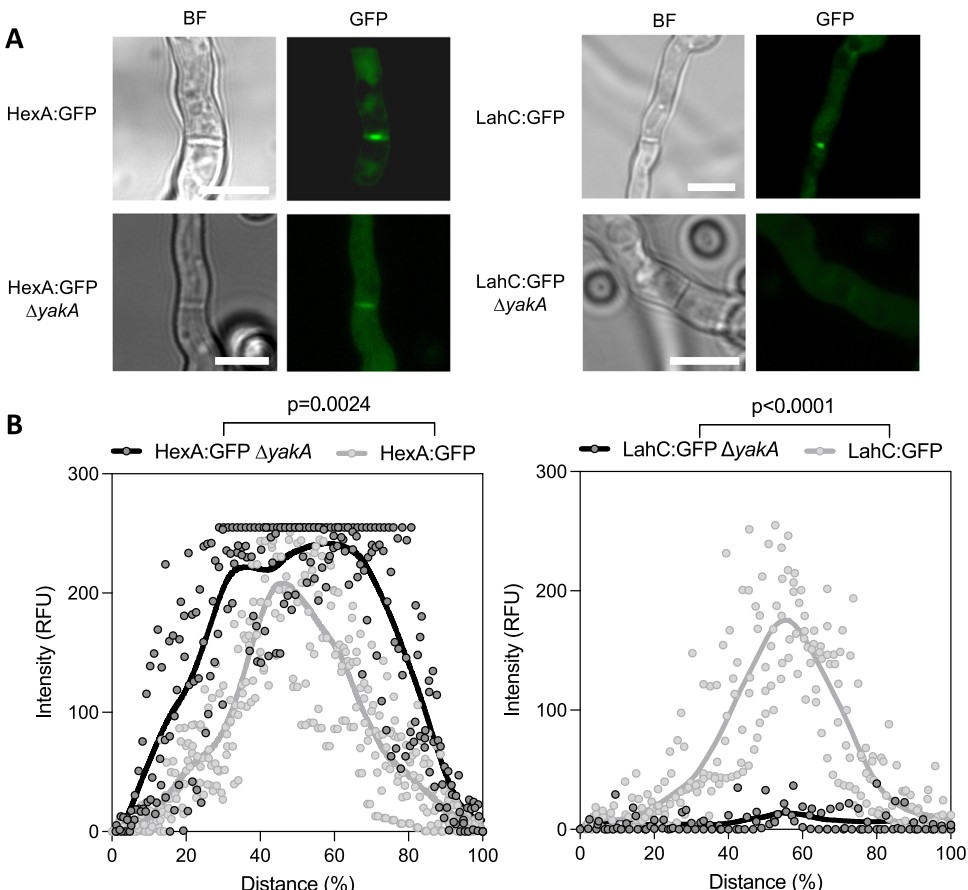

**Fig. 6 | Microscopic characterisation of septal pore components; HexA and Lah.**
**A** Microscopy of single hyphae of HexA-GFP and LacC-GFP upon iron deleted conditions show *yakA*-dependent localisation to the septum or the septal pore. Representative images of HexA-GFP Δ*yakA* and LahC-GFP Δ*yakA* showing dysregulation of HexA and Lah upon deletion of *yakA*. Scale bar = 10 μm. **B** Quantification of fluorescent intensity measured across the septum for HexA-GFP and LahC-GFP in the parental isolate or upon deletion of *yakA* (*n* = 3). Fluorescent intensity was normalised to the total hyphal width to account for variability. Statistical difference was assessed by two-way repeated measures ANOVA. Source data are provided as a Source Data file.

multinucleate hyphal compartments and heterokaryons formed through this process are likely to impact the effect of an individual mutation[48–50]. In *A. fumigatus* however, hyphal fusion occurs rarely in laboratory conditions unless strains are placed under stress and then only when cultured on solid media[28,48]. Taking advantage of this, and our ability to generate barcoded null mutants rapidly in *A. fumigatus*[23], we have developed an accessible approach to evaluate strain fitness in a collection of protein kinase null mutants. Hierarchical clustering of our data highlights the previously described interconnectivity of a number of protein kinases including the cell wall integrity (CWI) MAPK signalling pathway[29] and also reveals potential links between other regulators such as *halA* (AFUB_053500), an orthologue of the *S. cerevisiae* salt tolerance kinase HAL4 and *skyA* (AFUB_096590). While these associations do not necessarily represent direct interactions, it is possible that these kinases may regulate an overlapping cohort of proteins.

In this study we also demonstrate that competitive profiling can be applied to animal models of infection. We have shown that two protein kinases with known defects in virulence, namely *pkaC1* (AFUB_027890)[51], and the CWI signalling kinase *mkkA* (AFUB_006190)[52] have significant reductions in fitness in a leukopenic mouse model of infection. Our data is equally compelling for another member of the CWI-MAPKs cascade, MpkA, however this differs from published data on this protein kinase which indicates that *mpkA* is dispensable for virulence[29]. The Δ*bckA* mutant, which lacks the MAP3K member of the CWI-MAPK cascade, also showed a fitness defect in our

mouse model, alongside several other protein kinases including Δ*yakA* (ΔAFUB_099170). However, variability within experiments resulted in these reductions in virulence not reaching statistical significance. Interestingly, the distribution of fitness values for Δ*yakA* in the mouse model was bimodal (Supplementary Fig. 1), with some datapoints showing limited fitness defects and some with severe fitness defects. Our data could be indicating that the stress caused by exposure to the mouse lung is inducing hyphal fusion and hence the sharing of genetic material (and in turn the genetic barcodes that we use to assess fitness) which in turn would mask fitness defects in these strains[28]. Alternatively, as seen in our histological data from mice infected with the Δ*yakA* isolate, the data may indicate that growth of Δ*yakA* is discontinuously distributed throughout the various niches within the lung.

YakA is one of two DYRK kinases in *A. fumigatus*, however our data indicate that it has a non-redundant role in adaptation to pH stress, triazole stress, non-fermentable carbon sources, environments with physiological levels of iron, the ability to penetrate solid substrates and establish an infection in a mouse model of aspergillosis. In *S. cerevisiae*, the YakA orthologue, Yak1p is a core component of the glucose sensing system that is negatively regulated by TOR[53]. Upon activation, Yak1p is imported into the nucleus where it phosphorylates a suite of transcription factors including Pop2p, Msn2, Msn4 and Hsf1p which in turn initiate transcriptional adaptive stress responses[37,54]. Recently it has been suggested that Yak1 is localised to the nucleus and directly phosphorylates the iron homoeostasis transcription factor HapX to

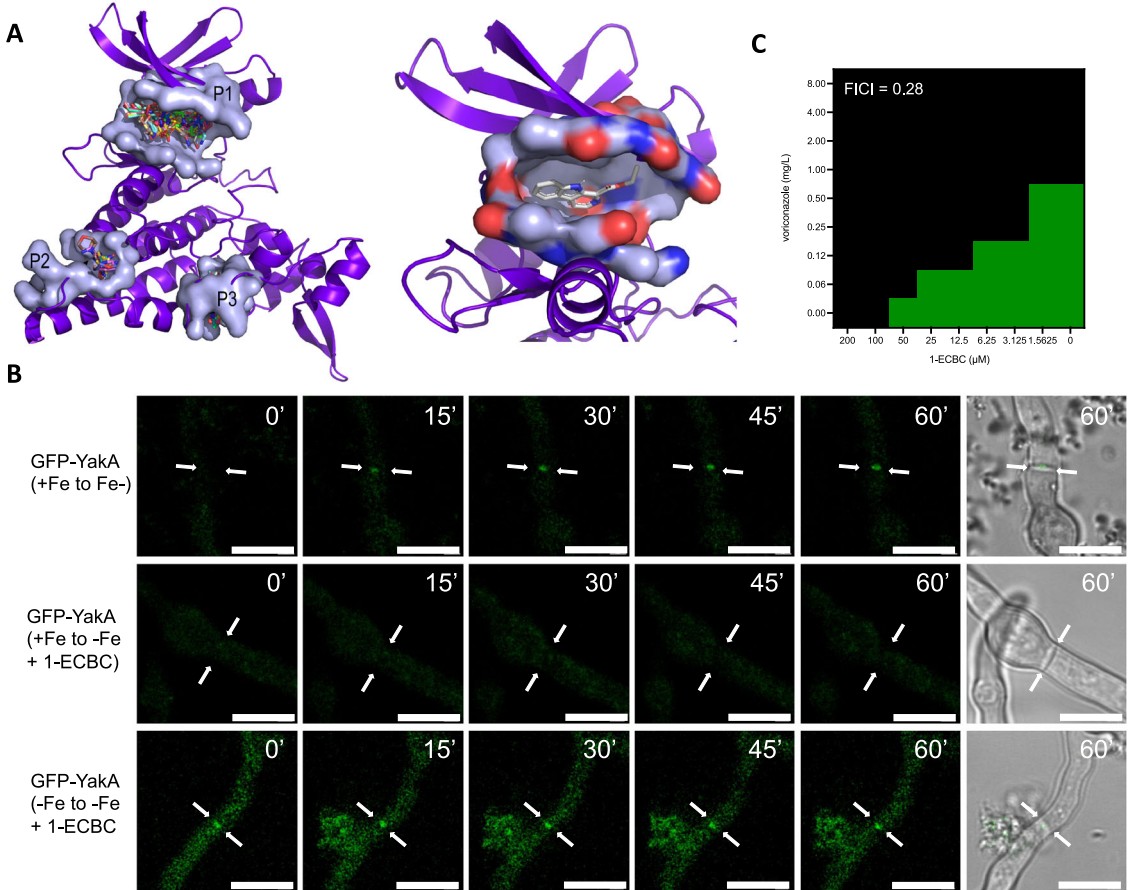

**Fig. 7 | YakA as a druggable target in *A. fumigatus*. A** An Alphafold2 model of the YakA protein docked to a library of small molecules using blind docking. Druggable pockets P1–P3 are shown as surface renders. The biggest pocket, P1, binds the largest number of fragments (778/1000), and is at the active site of the kinase. The druggability score from PockDrug for P1 is 0.81. Two smaller pockets, P2 and P3, are found on the C-lobe of the YakA. Druggability scores for P2 and P3 are 0.73 and 0.64 respectively. Pocket P1 is shown in further detail in complex with the 1-ECBC molecule. Charged residues in the pocket are indicated. The ΔG of 1-ECBC at the top scoring binding position is −7.7 kcal/mol. **B** Microscopic evaluation of YakA-GFP upon iron limitation and addition of 1-ECBC. Hyphae were followed for 1 h upon 1-ECBC addition. ECBC prevents localisation of YakA to the septal pore, but does not induce removal once it is in place. Scale bar = 10 μm. **C** Checkerboard assay (*n* = 3) for 1-ECBC and voriconazole to assess synergism of these drugs to inhibit growth of *A. fumigatus* MFIG001. The FICI (top-left) was calculated and shown synergism (<0.5).

facilitate adaptation to high iron levels in the filamentous fungus *Fusarium graminarum*[38]. Although our phosphoproteomic data indicates that HapX is phosphorylated (position 370-383), we saw no evidence that this was in response to YakA activity nor did we see any evidence of nuclear localisation of YakA in response to iron depletion, rather our data indicates that upon stress YakA localises to the septal pore. Consistent with this, we were unable to show a role for YakA in direct regulation of biosynthesis of the siderophore TAFC. Furthermore, although the phosphoproteomic profiles we observed revealed a key role for YakA in regulating 28% of the phosphopeptides observed in response to iron depletion, critically it was not involved in regulating the phosphorylation of proteins involved in siderophore biosynthesis or iron uptake (Supplementary Data 3).

*A. fumigatus* hyphae are considered to be largely syncytial, i.e. the content of the cytoplasm is near-uniformly distributed and readily exchangeable. However, the hyphae of filamentous fungi have cross walls, termed septa, that enable compartmentalisation of cellular material[55]. These septa have pores that permit streaming of cytoplasm and between compartments but these pores can be transiently blocked by an organelle known as the Woronin body[56]. It has been postulated that septal pore blocking permits an effective way of transiently controlling protein, nucleic acid and nutrient concentrations between compartments[57]. Notably, the phosphopeptides

dysregulated upon loss of *yakA* were heavily enriched from proteins associated with the septa. This aligned with our data that shows YakA localising to the septal pore in response to low iron stress. We also show that septal plugging by the Woronin body complex component HexA, and localisation of the C-terminal domain of the leashin Lah, a protein that tethers HexA to the septa, is YakA dependent and this localisation is also triggered in low iron environments. Seemingly therefore, YakA has been repurposed in *A. fumigatus* to regulate hyphal compartmentalisation upon stress. However, like Yak1 in *S. cerevisiae*, activation of YakA can be induced by rapamycin and hence appears to be under the control of TOR complex. Intriguingly, it has been demonstrated that the TOR kinase is a key regulator of iron homoeostasis, including the transcriptional regulator HapX[58]. Our current hypothesis therefore is that, under iron limitation TOR activity is reduced, resulting in a cascade that results in activation of siderophore production via HapX and initiation of septal blocking by the Woronin body, mediated by YakA to enrich the growing tip for iron.

The overall objective of this study was to identify novel targets for antifungal therapy that would enable the development of compounds that could synergise with the azoles. Serendipitously, as we were discovering the role of YakA in septal plugging, an inhibitor of *C. albicans* Yak1 was being described[44]. The *Lactobacillus* derived compound 1-acyl-β-carboline (1-ABC) was shown to prevent Yak1-mediated

filamentation of *Candida* species and inhibit Yak1 activity in vitro. Here we have shown that an analogue 1-ECBC is able to prevent low-iron induced septal plugging. Intriguingly however, once septa are blocked, chemical inhibition of YakA activity does not appear to unblock septal pores, indicating that unblocking septa either does not occur in low-iron conditions or that unblocking of septa is mediated by another, as yet unidentified protein. Consistent with our evidence, and that of others[59] that septal plugging is critical for adaptive response to triazole drug challenge, 1-ECBC synergises with voriconazole to inhibit growth of *A. fumigatus*, however given that 1-ECBC may have off target effects we cannot conclusively conclude that is this as a direct result of the compound on YakA activity. As we have shown that YakA is also required for virulence of *A. fumigatus*, there is a clear possibility that β-carbolines, which have been developed to target the YakA orthologue DYRK1A for use as an anti-depressant or a treatment for Parkinson's disease, could represent a novel class of clinical therapeutics that synergise with the azoles[60].

YakA inhibitors may also have value in crop protection. In the cereal pathogen *Fusarium graminearum*[38], the rice blast fungus *Magnaporthe oryzae*[34] and *Botrytis cinerea*[33], the causative agent of grey mould, Yak1 orthologues have been shown to be essential for virulence and in the case of *Magnaporthe oryzae*, required for the formation of a cell essential for pathogenesis known as the appressorium. It is noteworthy that many plants produce β-carboline alkaloids and it is interesting to speculate that production of such compounds may provide a natural defence against fungal pathogens[61].

To summarise, we have applied a functional genomic approach to identify targets for antifungal therapy. In doing so we have uncovered a role for the DYRK1 kinase YakA in the regulation of septal pore plugging. Chemical perturbation of YakA function using 1-ECBC enhances the action of the azole class of antifungals s opening the possibility that β-carbolines could be developed for clinical and/or agricultural purposes.

## Methods

### Ethics statement
The mouse infection experiments were performed under UK Home office Project Licence PDF8402B7 and approved by the University of Manchester Ethics Committee.

### In silico identification of kinases
Protein kinases from ORF translations for the Af293 genome were identified using Kinomer v1.0[24,62]. In Kinomer, a probability threshold of 0.001 was set to define a hit from this screening activity however all hits below this threshold were also evaluated using the Simple Modular Architecture Research Tool (SMART) to identify any protein kinase domains. A semantic text search to identify other previously annotated protein kinases in the genome of Af293 was performed using the term kinase at FungiDB[63]. Assessing the PFAM descriptors filtered out small molecule kinases and kinase interacting proteins[64]. PFAM domains PF02518, PF13598, PF07730 and PF07536 in the A1163 and Af293 genomes were used to identify histidine kinases. Kinases protein sequences were aligned using RaxML with default settings, and the resulting phylogenetic tree was viewed using Interactive Tree of Life (iToL)[65,66].

### Generation and validation of the A. fumigatus kinase knockout library
*A. fumigatus* MFIG001 was used as the parental isolate to generate kinase knockout mutants[24]. Knockout cassettes were generated using the protocol from Zhao et al.[23]. Briefly, 1 kb flanking regions of the gene of interest were PCR amplified and fused to a hygromycin resistance cassette via additional fusion PCR. *A. fumigatus* MFIG001 was cultured in Sabouraud agar (Oxoid, UK) overnight at 37 °C at 120 rpm, followed by protoplasting for 3 h in Sabouraud agar + VinoTaste Pro solution

(freshly filtered in 0.6 M KCl, Citric Acid). Protoplasts were filtered through Miracloth, washed twice in 0.6 M KCl and resuspended in 0.6 M KCl + 200 mM CaCl2. Fusion PCR product was added to $1 \times 10^5$ protoplasts, followed by addition of PEG. This was incubated on ice for 30 min. In total, 600 μl of PEG was added and the mixture was then incubated at room temperature for 10 min. Transformation mixture was plated on selective medium (YPS + 100 mg/l hygromycin). Transformants were twice purified on Sabouraud agar + 100 mg/l hygromycin and PCR validated.

The *yakA* null mutant was reconstituted to give the *yakA*+ isolate. This strain was generated by using selection-free CRISPR-Cas9 mediated transformation[29]. Briefly, the *yakA* gene along with c.1200 bp of flanking DNA was PCR amplified using primers AFUB_099170P1-AFUB_099170P4 and genomic DNA from strain MFIG001 as a template. This PCR product was used as a repair template to transform the ΔyakA strain. Transformants were picked and tested for susceptibility to hygromycin on Sabouraud agar + 100 mg/l hygromycin and replicate plated on Sabouraud agar. Transformants sensitive to hygromycin were selected from the Sabouraud agar plate and PCR validated using AFUB_099170P1- AFUB_099170P4 and internal hygromycin (Hyg_Fw and Hyg_Rv) primers (Supplementary Data 1).

### Promoter replacement
Kinase knockout mutants that could not be generated after three transformation attempts, were considered potentially essential. Repair template was generated by PCR, using the tetOFF pSK606[67] as template and primers (Supplementary Data 1) for mutants. This repair template and crRNAs (Integrated DNA Technologies) (Supplementary Data 1) were used for CRISPR-Cas9 transformations to perform promoter replacement[68]. Briefly, crRNAs were designed to target the translational start site (ATG) of each individual gene. The closest PAM site with scores above 0.5, as calculated by EuPaGDT, was used, and 20 bp crRNA were ordered from Integrated DNA Technologies, UK. CRISPR/Cas9 mediated transformation was carried out as previously described. Briefly, *A. fumigatus* was grown overnight at 37 °C in Sabouraud agar, followed by protoplasting using Vinotaste. Protoplasts were washed twice in 0.6 M KCl, followed by resuspension in 0.6 KCl + 200 mM CaCl2. Guide RNA was formed by annealing crRNA to tracrRNA (IDT) and RNPs were formed by incubating at room temperature for 5 min with purified SpCas9 (IDT). Repair template was generated by performing PCR with 50-bp microhomology arms included in the oligos (Supplementary Data 1). Repair template, RNPs and protoplasts were mixed with PEG-CaCl$_2$, and incubated at 50 min on ice. In total, 600 μl of PEG-CaCl2 was added, followed by incubation at room temperature for 20 min. Protoplasts were spread onto AMM + 1% Sorbitol + 100 mg/l pyrithiamine, left at room temperature for 1 day, followed by incubation at 37 °C for 5 days. Transformants were purified twice on AMM + 100 mg/l pyrithiamine and validated by PCR.

### Competitive fitness profiling using bar-seq
Unless otherwise stated, *A. fumigatus* strains used in this study were cultured on Sabouraud Dextrose agar (Oxoid) for 3 days at 37 °C. Conidia were harvested in phosphate buffered saline + 0.01% Tween-20 (PBS-T) and collected by filtration through Miracloth (Millipore Limited cat. no. 475855). Conidia were quantified using a Fuchs-Rosenthal haemocytometer.

Conidia from each mutant was harvested and quantified using a haemocytometer, then normalised to $5 \times 10^8$ cell/ml and pooled into one tube. For in vitro competitive fitness assay, 100 μl of the pool of kinase mutants was added to each flask containing 50 ml of fungal RPMI (fRPMI) media[69] to achieve a final conidial concentration of $1 \times 10^6$ cells/ml. The experimental conditions included oxidative stress (1.5 mM H$_2$O$_2$), pH stress (pH4 and pH8), temperature stress (48 °C and 30 °C), antifungal stress (voriconazole 0.15 mg/l, olorofim 0.002 mg/l)

and different iron concentrations (Fe-, BPS, 30 μM, 300 μM, 5 mM). All conditions were performed in triplicate and incubated in a shaking incubator at 180 rpm for 20 h at 37 °C, apart from the high (48 °C) and low temperature (30 °C) conditions.

After incubation, the flasks were removed from the incubator and the biomass from each flask was collected using Miracloth® and a Büchner funnel under vacuum. Liquid nitrogen was used to snap freeze the dried biomass and ground into a powder, using a sterile mortar and pestle. 0.1 g from each baffled flask was collected for DNA extraction.

Total fungal DNA was extracted using a standardised CTAB DNA extraction[70], for tissue samples an additional phenol:chloroform cycle was carried out. Enrichment PCR was performed using Phusion U polymerase and enrichment primers (Supplementary Data 1) for 1 cycle at 98 °C for 30 s, followed by 30 cycles at 98 °C for 10 s, 65 °C for 30 s, 72 °C for 30 s, followed by a final extension step at 72 °C for 10 min. Enriched barcodes were cleaned using AMPure Beads and indexed with the Nextera XT kit (NEB) following manufacturers protocol. Indexed products were cleaned using AMPure Beads at a 1:1 ratio. Sequencing was performed on Illumina iSeq following manufacturers protocol.

Quality control of raw reads was performed using FastQC (0.12.1), and trimmed using CutAdapt (4.9). Trimmed reads were aligned to a Fasta file containing individual barcodes using Bowtie2 (2.5.0). Counts per strain were obtained using BED IdXstats. Fitness indices were calculated using DEseq2 (1.38.3). Clustering was performed using Pheatmap (1.0.12) in Rstudio (4.2.2) with standard parameters.

## Phenotypic and susceptibility analysis

$10^3$ spores of MFIG001, Δ*yakA* and Δ*yakA::yakA*⁺ strains were inoculated onto solid iron deficient minimal medium (MM) [1% (w/v) glucose, Ammonium Tartrate 5 mM, MgSO₄ 4.3 mM, KCl 7 mM, KH₂PO₄ 11 mM, trace elements without iron, pH 6.5] plates with increasing concentrations of agarose (0.5 to 6%) and FeSO₄ (0 to 300 μM). Trace elements composition was as described in Käfer[36]. Plates were incubated for 48 or 72 h at 37 °C and measurements or images were taken.

Susceptibility testing was performed according to EUCAST methodology in RPMI-1640 medium MOPS buffered with 2% glucose. Growth was observed microscopically after 48 h according to guidelines for *A. fumigatus*. FICI was calculated by dividing MIC under combinatorial conditions by the MIC to single compounds.

## Galleria mellonella and mouse infection models

The *Galleria mellonella* infection model was performed as described in Johns et al.[71]. Briefly larval *G. mellonella* (Live Foods Company (Sheffield, England)) (Minimum weight 0.3 g; 10 per group) were inoculated by injecting 10 μl of a $1 \times 10^6$ spores/ml suspension into the last, left proleg. Sham infections were performed with PBS. Larvae were monitored daily for 8 days and scored for mortality.

The mouse infection experiments were performed under UK Home office Project Licence PDF8402B7 and approved by the University of Manchester Ethics Committee. *A. fumigatus* strains were cultured on ACM containing 5 mM ammonium tartrate for 6 days at 37 °C, and conidia were harvested in sterile saline and used for infection experiments. SH-*aft4* was included to allow to control for the potential effect of introducing a hygromycin selection marker[72]. Ten week old CD1 male mice (27–37 g) (Charles River UK, Ltd.) were housed in groups of 3–4 in IVC cages with access to food and water ad libitum. Mice were maintained in a 12 h light:dark cycle, at 21 °C plus or minus 2 degrees. Relative humidity was maintained between 45 and 65%. All mice were given 2 g/l neomycin sulfate in their drinking water throughout the course of the study. For the leukopenic model of infection, mice were rendered leukopenic by administration of cyclophosphamide (150 mg/kg of body weight; intraperitoneal) on days −3, −1, +2 and a single subcutaneous dose of triamcinolone (40 mg/kg) was

administrated on day −1. Mice were anaesthetised by exposure to 2–3% inhalational isoflurane and challenged by intranasally with a conidial suspension of $1.25 \times 10^7$ conidia/ml in 40 μl of saline solution. Mice were weighed every 24 h from day −3, relative to the day of infection, and visual inspections were made twice daily. In the majority of cases, the endpoint for survival in experimentation was a 20% reduction in body weight measured from day of infection, at which point mice were culled. Kaplan–Meier survival analysis was used to create a population survival curve and to estimate survival over time, and *p* values were calculated through a log rank analysis.

## Histology analysis

Immunosuppressed male CD1 mice (*n* = 3) were infected as described above. After 36 h of infection, mice were culled. Lungs for histological analysis were immediately fixed in 4.0% (v/v) paraformaldehyde (Sigma-Aldrich), and subsequently embedded in paraffin. Four-micrometre sections were stained with haematoxylin–eosin (H&E) and Grocott's Methenamine Silver (G.M.S). Images were taken using a Pannoramic 250 Flash Slide Scanner (3D HISTECH) using brightfield illumination. Analysis was performed in ImageJ.

## Phosphoproteomics

Strain MFIG001 and Δ*yakA* were cultivated in AMM with iron-free Hutner's trace element solution under −Fe conditions or under +Fe conditions (addition of 30 μM FeSO₄) in 500 ml flasks for 18 h, 200 rpm at 37 °C. Mycelia were harvested using Miracloth (Merck Millipore, Germany) and disrupted by using mortar and pestle with liquid nitrogen. Cell debris were homogenised in lysis buffer (1% (w/v) SDS, 150 mM NaCl, 100 mM TEAB (triethyl ammonium bicarbonate), one tablet each of cOmplete Ultra Protease Inhibitor Cocktail and Phos-STOP). After addition of 0.5 μl Benzonase nuclease (250 U/μl) the samples were incubated at 37 °C in a water bath sonicator for 30 min. Proteins were separated from unsolubilised debris by centrifugation (15 min, 18,000 × *g*). Each 6 mg of total protein per sample was diluted with 100 mM TEAB to gain a final volume of 4 ml. Subsequently, cysteine thiols were reduced and carbamidomethylated in one step for 30 min at 70 °C by addition of 120 μl of 500 mM TCEP (tris(2-carboxyethyl)phosphine) and 120 μl of 625 mM 2-chloroacetamide (CAA). The samples were further cleaned up by precipitation with 20% trichloroacetic acid (TCA) for 30 min on ice. After centrifugation (20 min at 20,000 × *g*), the precipitate was washed with 90% acetone. Protein precipitates were resolubilized in 5% trifluoroethanol of aqueous 100 mM TEAB and digested overnight (18 h) with a Trypsin + LysC mixture (Promega) at a protein to protease ratio of 25:1. An aliquot of 0.2 mg digested protein was used for the reference proteome analysis and 5.8 mg was used for the phosphopeptide enrichment. Samples were evaporated in a SpeedVac. The reference proteome sample was resolubilized in 50 μl of 0.05% TFA in H2O/ACN 98/2 (v/v) filtered through PES 10 kDa MWCO membrane spin filters (VWR). The filtrate was transferred to HPLC vials and injected into the LC-MS/MS instrument.

Phosphopeptides were enriched by using TiO₂ + ZrO₂ TopTips (Glygen Corp., Columbia, MD, USA). TopTips were loaded with 500 μg protein isolate using 12 TopTips per biological replicate after equilibration with 200 μl Load and Wash Solution 1, LWS1 (1% trifluoroacetic acid (TFA), 20% lactic acid, 25% acetonitrile (ACN), 54% H2O). TopTips were centrifuged at 200 × *g* for 5 min at room temperature. After washing with 200 μl LWS1, the TiO₂/ZrO₂ resin was washed with 25% ACN and subsequently the phosphopeptides were eluted with 200 μl NH₃· H₂O (NH₄OH), pH 12. The alkaline solution was immediately evaporated using a SpeedVac. The phosphoproteome samples were resolubilized in 50 μl of 0.05% TFA in H₂O/ACN 98/2 (v/v) filtered through PES 10 kDa MWCO membrane spin filters (VWR). The filtrate was also transferred to HPLC vials and injected into the LC-MS/MS instrument.

Each sample was measured in triplicate (3 analytical replicates of 3 biological replicates of a reference proteome fraction and a phosphoproteome fraction. LC-MS/MS analysis was performed on an Ultimate 3000 nano RSLC system connected to a QExactive HF mass spectrometer (both Thermo Fisher Scientific, Waltham, MA, USA). Peptide trapping for 5 min on an Acclaim Pep Map 100 column (2 cm × 75 μm, 3 μm) at 5 μl/min was followed by separation on an analytical Acclaim Pep Map RSLC nano column (50 cm × 75 μm, 2 μm). Mobile phase gradient elution of eluent A (0.1% (v/v) formic acid in water) mixed with eluent B (0.1% (v/v) formic acid in 90/10 acetonitrile/water) was performed using the following gradient for the more hydrophilic phosphoproteome samples: 0–5 min at 4% B, 15 min at 7% B, 50 min at 10% B, 100 min at 14% B, 150 min at 25% B, 190 min at 60% B, 205–215 min at 96% B, 215.1–240 min at 4% B. The reference proteome gradient was as follows: 0–4 min at 4% B, 10 min at 7% B, 50 min at 12% B, 100 min at 16% B, 150 min at 25% B, 175 min at 35% B, 200 min at 60 %B, 210–215 min at 96% B, 215.1–240 min at 4% B.

Positively charged ions were generated at spray voltage of 2.2 kV using a stainless steel emitter attached to the Nanospray Flex Ion Source (Thermo Fisher Scientific). The quadrupole/orbitrap instrument was operated in Full MS/data-dependent MS2 Top15 mode. Precursor ions were monitored at $m/z$ 300–1500 at a resolution of 120,000 FWHM (full width at half maximum) using a maximum injection time (ITmax) of 120 ms and an AGC (automatic gain control) target of $3 \times 10^6$. Precursor ions with a charge state of $z = 2$–5 were filtered at an isolation width of $m/z$ 1.6 amu for further HCD fragmentation at 27% normalised collision energy (NCE). MS2 ions were scanned at 15,000 FWHM (ITmax = 100 ms, AGC = $2 \times 10^5$) using a fixed first mass of $m/z$ 120 amu. Dynamic exclusion of precursor ions was set to 30 s and the minimum AGC target for Precursor ions selected for HCD fragmentation was set to $1 \times 10^3$. The LC-MS/MS instrument was controlled by Chromeleon 7.2, QExactive HF Tune 2.8 and Xcalibur 4.0 software.

Tandem mass spectra were searched against the FungiDB database (2021/10/26 (YYYY/MM/DD); https://fungidb.org/common/downloads/Current_Release/AfumigatusAf293/fasta/data/FungiDB-54_AfumigatusAf293_AnnotatedProteins.fasta) of *Aspergillus fumigatus* Af293 using Proteome Discoverer (PD) 2.4 (Thermo) and the algorithms of Mascot 2.4.1 (Matrix Science, UK), Sequest HT (version of PD2.4), MS Amanda 2.0, and MS Fragger 3.2. Two missed cleavages were allowed for the tryptic digestion. The precursor mass tolerance was set to 10 ppm and the fragment mass tolerance was set to 0.02 Da. Modifications were defined as dynamic Met oxidation, phosphorylation of Ser, Thr, and Tyr, protein N-term acetylation and/ or loss of methionine, as well as static Cys carbamidomethylation. A strict false discovery rate (FDR) < 1% (peptide and protein level) and a search engine score of >30 (Mascot), >4 (Sequest HT), >300 (MS Amanda) or >8 (MS Fragger) were required for positive protein hits. The Percolator node of PD2.4 and a reverse decoy database was used for $q$ value validation of spectral matches. Only rank 1 proteins and peptides of the top scored proteins were counted. Label-free protein quantification was based on the Minora algorithm of PD2.4 using the precursor abundance based on intensity and a signal-to-noise ratio >5. Normalisation was performed by using the total peptide amount method. Imputation of missing quan values was applied by using abundance values of 75% of the lowest abundance identified per sample. For the reference proteome analysis used for master protein abundance correction of the phosphoproteome data, phosphopeptides were excluded from quantification. Differential protein abundance was defined as a fold change of >2, $p$ value/lABS(log4ratio, <0.05 and at least identified in 2 of 3 replicates. Differential phosphopeptide abundance was defined as a fold change of >2, ratio-adjusted $p$ value < 0.05 $p$ value/ABS(log4ratio) and at least identified in 2 of 3 replicates.

## Microscopic analysis
The LahC:GFP and HexA:GFP were obtained from Frank Ebel[25,73]. For live-cell imaging in liquid cultures, conidia at concentration of $5 \times 10^5$ cells/ml were dispensed into the wells of an 8-well ibidi imaging chamber (ibidi GmbH, Martinsried, Germany), and incubated in 250 μl Hutner's minimum media (HMM) for 20 h at 37 °C for the development of matured hyphae before imaging. The imaging chamber was filled with medium (either iron depleted or replete HMM) dependent on the needs of each individual experiment. For medium shifting experiments, 250 μl of the extant culture medium was removed from each well of the imaging chamber and replaced with desired medium (either iron repleted HMM, iron depleted HMM or iron depleted HMM with different reagents) by pipetting. Reagents used to supplement medium included rapamycin (10 μM), $H_2O_2$ (3 mM), caspofungin (0.03 mg/l), voriconazole (1.5 mg/l) or 1-ECBC (500 μM) or 1.8 M glycerol. Fungal cells were imaged for periods of up to 1 h at 37 °C in a temperature-controlled chamber mounted on the microscope stage.

For live-cell imaging of fungal penetration on solid medium, 100 conidia were spotted on the surface of HMM media containing 3% or 4% of agarose. The petri dishes containing inoculated media were wrapped in Parafilm and incubated at 37 °C for 24 h. Before imaging, a cubic of the media (-1 cm × 1 cm) with the fungal colony in the centre was cut out using a sterile scalpel, and placed into a well of an imaging chamber facing downwards. The well was prefilled with 100 μl of distilled water.

Live-cell imaging of *A. fumigatus* in liquid cultures was performed using a Leica TCS SP8 confocal laser-scanning microscope (Leica Microsystems Ltd., Milton Keynes, UK) equipped with photomultiplier tubes, hybrid GaAsP detectors and a 63x water immersion objective. Excitation and emission wavelength of 496 nm and 515–545 nm respectively were used for imaging the GFP expressed by *A. fumigatus*.

For live-cell imaging of fungal penetration on solid medium, a long working distance 25x water immersion objective lens was employed instead. The 'Z-compensation function' was used for fine adjustment of the laser excitation to compensate the signal reduction in the deeper section of the solid media. These adjustments were done across 3 points of the z-stack, preventing over or under-exposure of acquisition. Acquired images were analysed using Imaris v8.0 software (Bitplane Scientific software module; Zurich, Switzerland).

## In silico small molecule screening
The structure of YakA kinase domain (Y345-I676), as predicted by ScanProsite[74], was determined using AlphaFold2[75], for subsequent analysis, the highest-scoring model was used (pLDDT: 96.1, pTMscore: 0.9375). VSpipe (1.0)[43], a semi-automated pipeline, was used for blind docking with the Maybridge Ro3 1000 fragment library using AutoDock Vina (1.1.2)[76]. This library contains 1000 chemically diverse, rule of three (Ro3) compliant, and pharmacophore-rich fragments. Clusters were defined as having at least 15 fragments bind to it. 1-ECBC was also used for docking using AutoDock Vina. Druggable pockets were predicted using PockDrug[77], an online server which assess pocket geometry, hydrophobicity, and aromaticity and gives a druggability score where scores of >0.5 are considered druggable. Outputs from all programmes were visualised in PyMOL 2.5.

## Statistics and reproducibility
All experiments were performed three independent times unless stated otherwise. Statistical analyses were performed in GraphPad PRISM 8.0.1 (La Jolla, CA, USA) using one-way ANOVA, log rank Mantel−Cox test and two ANOVA, as indicated in the figure legends. No data were excluded from the analyses.

## Reporting summary
Further information on research design is available in the Nature Portfolio Reporting Summary linked to this article.

## Data availability

Mass spectrometry proteomics data are available from the ProteomeXchange Consortium via the PRIDE partner repository with dataset identifier PXD042616. Source data are provided with this paper.

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

## Acknowledgements

We thank Silke Steinbach for technical support and Robb Cramer and Jessie MacAlpine for helpful discussions around the function of YakA. We would also like to thank Leah Cowen for sending us the 1-ECBC compound. This work was funded by the Wellcome Trust under project numbers 208396/Z/17/Z (to M.J.B., P.B. and D.D.) and 219551/Z/19/Z (to M.J.B.). Additional funds were received from the Deutsche Forschungsgemeinschaft (DFG, German Research Foundation) Collaborative Research Centre/Transregio FungiNet 124 'Pathogentic fungi and their human host: Networks of Interaction' (project number 210879364; project A1 and Z2) (to A.A.B.) and the International Leibniz Research School (ILRS) for Microbial and Biomolecular Interactions as part of the excellence graduate school Jena School for Microbial Communication (JSMC) (to A.A.B.). We thank Fundação de Amparo à Pesquisa do Estado de São Paulo (FAPESP) 2021/04977-5 (to G.H.G.) and 2020/01131-5 and 2018/00715-3 (to C.V.) and Conselho Nacional de Desenvolvimento Científico e Tecnológico (CNPq) 301058/2019-9 and 404735/2018-5 (to G.H.G.), both from Brazil, and also a National Institutes of Health National Institute of Allergy and Infectious Diseases grant (R01AI153356) from the United States (to G.H.G.). *The Ministry of Higher Education and Scientific Research (MOHESR), Iraq.* to N.A.F and M.B.

## Author contributions

Conceptualisation: M.J.B. Methodology: N.v.R., C.Z., C.V., O.K., T.K., E.B., H.H., and M.J.B. Software: H.C., N.v.R., and M.J.B. Validation: N.v.R., C.Z., N.A.F., C.V., and H.B.S. Formal analysis: N.v.R., C.Z., N.A.F., I.S.R.S., C.V., S.G., H.C., C.B., R.F.G., H.B.S., L.I., O.K., T.K., E.B., D.D., P.B., H.H., and M.J.B. Investigation: N.v.R., C.Z., N.A.F., I.S.R.S., C.V., S.G., H.C., C.B., R.F.G., H.B.S., L.I., O.K., T.K., and J.A. Data curation: N.v.R., C.Z., N.A.F., T.K., and M.J.B. Writing—original draft: N.v.R. and M.J.B. Writing—review and editing: all authors. Visualisation: N.v.R., C.Z., I.S.R.S., C.V., and C.B. Supervision: O.K., E.B., G.H.G., J.A., D.D., A.A.B., H.H., and M.J.B. Funding acquisition: H.H., C.V., G.H.G., A.A.B., P.B., D.D., A.A.B., and M.J.B.

## Competing interests

The authors declare no competing interests.

## Additional information

[1]Manchester Fungal Infection Group, Division of Evolution, Infection and Genomic Sciences, Faculty of Biology, Medicine and Health, University of Manchester, Manchester, UK. [2]Department of Pharmacology, College of Medicine, University of Kerbala, Kerbala, Iraq. [3]Division of Molecular Biology, Biocenter, Innsbruck Medical University, Innsbruck, Austria. [4]Department of Clinical Laboratory Sciences, College of Applied Medical Sciences, King Saud Bin Abdulaziz University for Health Sciences, 11481 Riyadh, Saudi Arabia. [5]Department of Molecular and Applied Microbiology, Leibniz Institute for Natural Product Research and Infection Biology (Leibniz-HKI), Jena, Germany. [6]MRC Centre for Medical Mycology, University of Exeter, Stocker Road, Exeter EX4 4QD, UK. [7]Faculdade de Ciências Farmacêuticas de Ribeirão Preto, Universidade de São Paulo, Ribeirão Preto, São Paulo, Brazil. [8]Mycology Reference Laboratory (Laboratorio de Referencia e Investigación en Micología [LRIM]), National Centre for Microbiology, Instituto de Salud Carlos III (ISCIII), Majadahonda, Madrid, Spain. [9]Division of Evolution, Infection and Genomic Sciences, Faculty of Biology, Medicine and Health, University of Manchester, Manchester, UK. [10]Department of Microbiology and Molecular Biology, Institute of Microbiology, Friedrich Schiller University, Jena, Germany. [11]These authors contributed equally: Norman van Rhijn, Can Zhao, Narjes Al-Furaiji. ✉e-mail: Mike.Bromley@manchester.ac.uk

