## [Peer Review File · Nature Communications]

Functional analysis of the *Aspergillus fumigatus* kinome identifies a druggable DYRK kinase that regulates septal pluggingReviewer #1 (Remarks to the Author):

This study focuses on the identification of *Aspergillus fumigatus* protein kinases that regulate disease-relevant phenotypes in an attempt to uncover novel antifungal drug targets. The disease burden associated with invasive fungal infections is high and treatment outcomes are poor. *A. fumigatus* is the most common cause of deeply invasive mold infections. Currently available antifungal drugs are limited and their continued use is further threatened by recent increases in resistance seen around the globe. For environmental fungi, like *A. fumigatus*, this resistance to the leading class of antifungals (triazoles) is partly driven by agricultural use of azole compounds. The authors report the generation of a barcoded *A. fumigatus* protein kinase deletion library that was subsequently screened in competitive fitness assays by bar-seq. Conditions used for fitness tests either mimicked stresses encountered in the host, or were actually performed in a mouse model. Although several kinases are found to support pathogenic and antifungal fitness, the authors focus on the YakA kinase. Data provided show that the YakA kinase is important for virulence in a mouse model of aspergillosis, characterized by reduced tissue invasion. Data also shows YakA is required for hyphal penetration into high percentage agar; loss of YakA causes reduced phosphorylation of septum-associated proteins; YakA localizes to the septal pore upon low iron stress where it subsequently promotes localization and activation of septal pore-plugging machinery. As the yakA deletion is also more susceptible to triazoles, the authors provide evidence that a known inhibitor of a YakA ortholog in yeast blocks septal localization of YakA and synergizes with triazoles in vitro. This work is significant for the field, as generation of the mutant library and successful employment in fitness screening could lead to additional finding down the road. This is especially important as novel antifungal targets that might both inhibit virulence but also synergize with current antifungals, are badly needed. Furthermore, protein kinases are readily targeted. The manuscript is fairly well-written, however, there were multiple omissions that have to be addressed. The figures are presented in a clear and logical manner, but the legends largely lack useful detail. Stats seem to be appropriately run. The conclusions are largely supported the data (with exception of issues noted below).

1. In general, the Figure Legends could use a bit more detail to keep the reader from continually referencing the MM section for things like stats and replicates.
2. Stats are not included on the survival graphs in Figure 2
3. There is no description of how the yakA reconstituted strain was generated.
4. It is unclear what is meant by the text in lines 540-543? No data is shown to support this vague statement but the authors use it to argue that YakA function in *A. fumigatus* is divergent from yeast.
5. For Figure 6, the difference in localization (focal point at septal pore vs wider localization over septum) is really evident by eye. How many replicates were measured here? The quantitative data only appears to show one septum per condition.
6. Does loss of yakA actual shift the MIC to triazoles based on EUCAST or CLSI guidelines? The growth assays are fine, but are on solid agar and show minor shift under non-standard conditions. Therefore, they do not correlate well with the checkerboard studies in Figure 7.
7. There is no description of the conditions under which the checkerboard assay was run.
8. Glucose uptake experiments are described in the methods but are found no where in the presented data.
9. What is Supplemental Figure 8A? It is not described.
10. Lines 765 and 784: Do the authors mean 1-ECBC?
11. The conclusion that the 1-ECBC/voriconazole synergism is due to blocking YakA localization to the septal pore is not well-supported. 1-ECBC is very likely to have off-target effects in the cell and the study lacks mutants that might help resolve this (YakA overexpression or hyperactive strains, etc.). This is not even offered as a caveat in the Discussion.
12. In this line of thought (#11), do aseptate mutants show the same hypersusceptibility to triazoles? These were previously published to not display triazole phenotypes (by CLSI MIC) in a previous study using a protein kinase disruption library (referenced by the authors). However, the authors are working in a different strain background. If these are in the authors collection and showed the same phenotype(s), it would be very supportive that it is actually the blocking of septal plugging by 1-ECBC treatment or loss of yakA that generates the triazole susceptibility phenotype.

Reviewer #2 (Remarks to the Author):

After deletion analysis of protein kinase genes in *Aspergillus fumigatus* a potential novel antifungal drug target was identified. Using competitive fitness profiling the potential target is shown to enhance sensitivity to azoles and has defects in fitness within models of infection. The target, yakA, locates and plays roles at septal pores and is involved in septal pore plugging and the ability of cells to penetrate solid media and grow in mice lung tissues. A yak1 inhibitory drug is shown to also affect septal plugging and synergize with azoles to inhibit *A. fumigatus* growth. Proteomic analysis identifies some potential targets of the kinase including septal pore proteins consistent with yakA potentially regulating septal pore functions. Overall, there are some important findings presented in this study and the data are overall clear and conclusions reached well-reasoned, with a few exceptions as detail further below. The manuscript could be improved by making the overall presented work easier to understand and with the many minor defects listed below considered.

More important issues:

L 49. Further data would need to be collected to conclude definitively that YakA regulates blocking of the septal pore upon exposure to stress via phosphorylation of the Woronin body tethering protein Lah. At the moment only a correlation is shown. Alternatively, the conclusions should be softened to not claim causation.

L 451. I could not find Supplementary Table 1. Assume what is meant is Supplemental Data 1, Library sheet. It would be more reader friendly to give better headings for all the Supplemental Data. For example, there are 5 sheets in Suppl data 1, none of which are labeled within the text. Similarly, there are 22 sheets in Supplemental Data 3 none of which are specifically mentioned in the text, all are referred to as Supplemental Data 3. The supplemental data should also be further reviewed as color designations are not explained and column headings are missing or difficult to understand.

L 451. Was there a reason not to use heterokaryon rescue to confirm that lack of recovery after gene deletion was actually caused because the target genes were essential, (PMID: 19883779 for example) rather than some other more mundane technical reason? There are methods to demonstrate heterokaryons are formed when an essential gene is deleted carrying both nuclei with the WT allele and nuclei with the null allele and also define the phenotype caused by the null mutation.

Suppl. Fig. 5B. The effect of agarose level on Δ yakA is not fully explained by saying it is exacerbated under iron limitation. There seems to be a graded, yet reversible, effect which is intriguing but not discussed. Does the fact that inhibition appears to be equally high at the lowest and highest levels of agarose fit with the model of turgor pressure issues?

L 626. Not sure it is possible to conclude that HexA depends upon YakA for location at the septal pore, it is there without yakA (Fig. 6). Further, no mention is made of the distribution pattern across the septum showing a central peak and two shoulder peaks. What are the shoulder peaks? This pattern is not discernable from the images presented and should be shown. Is the quantitation from a single imaging section or from a projection, same question for all the images shown. The peak likely represents the actual septal pore region (6B) and would suggest that there is increased HapA-GFP around this region without YakA, not a decrease, as concluded.

Because HexA-GFP locates to septal pore regions in low iron (6A) with or without yakA, how can it be concluded that YakA activity is required for micronutrient stress-initiated localization of Lah and HexA at the septal pore and subsequent septal plugging?

L 693. Several things in this paragraph are unclear. What is the difference in the leukopenic murine model of infection vs neutropenic mouse infection model vs murine model? L 697, what data is equally compelling? L 703, It is unclear what ...distribution of fitness values for replicates in the mouse model was bimodal... means? How can this be explained by induced hyphal fusion? Further details should be given for this argument.

Figure 7B. In the bottom panels, after shift from low iron to low iron + ECBC, it appears that the cell lyses at 15'. Is this a common event and is it dependent on the drug or due to the media shift. Would the effects be different if media exchange was done without added drug? This would seem to be an important control.

L 646. "Upon iron limitation 1-ECBC blocks YakA recruitment to the septal pore, which in turn blocks LahC recruitment (Figure 7B)." No data is apparent regarding the effects of 1-ECBC on LahC. Deletion of Yak1 and stopping its location to septa using a drug are two separate things.

Other issues:

L 35. Is *A. fumigatus* a pathogen or an opportunistic one?

Fig 1 A. Grey circles not obviously grey, too dark.

L 41. ...120 genetically barcoded null mutants.... L 102. ...108 genetically barcoded protein kinase null mutants... This discrepancy in the number of mutants should be explained for clarity.

L 404. Unclear what is meant by, ..now defunct annotations of the *A. fumigatus* genome..

Was there any correlation with the essential protein kinase genes previously defined in *A. nidulans* to those predicted in *A. fumigatus*?

L 456. Do the authors have an explanation for this discrepancy in their ability to obtain some null allele strains? Might need some further clarification/explanation.

Fig. 2A. 2A is somewhat impenetrable. The gene designations are difficult to read, gene names are missing and designations are not searchable. It is unclear how the competitive fitness growth experiments were completed. Were all strains inoculated together at one time?

L 1086. Wile should be while

Suppl Fig 2. What is FusC?

L 523. Yak1 has also been described and deleted in *A. nidulans*.

Suppl. Fig. 3. Pom1/2 is not mentioned in the manuscript main text.

L 587. Should it be Suppl Data 3?

Fig. 5B 6B. Is the scale correct (μm)? If so, these would be unusually wide cells that happened to be 100 μm wide. Also, for the graphs of the GFP signals at septa, what is the n and was any statistical analysis completed?

Size bars should be added to all cell image figures. At the moment, some are missing and some way too small to see, assuming the tiny bars in some images are in fact size bars?

L 632. Is adaptation the correct word here?

L642. What does figure refer to in (ΔG : -7.7 kcal/mol; figure)?

L 1078. Is shown the correct word here?

L 1129. GF-HexA needs correcting.

Supplemental Figure 8, nothing indicated regarding 8A.

L 646. Text regarding Supplemental Figure 8A does not match what is shown in the actual figure.

Some quantitation of data shown in Fig. 7B could be given.

L 647. ... which is turn... should be ..which in turn..

L 649. "... unblocking of the septal pore is a YakA independent process..." Not sure how this argument can be made when no unblocking of septal pores is shown.

FICI should be defined.

L. 667. Seems to not fit in this sentence.

L 677. Mixing of nuclei between cells should also be mentioned.

L 785. Should the drug not be 1-ECBC rather than 1-ABC?

Reviewer #3 (Remarks to the Author):

In this manuscript van Rhijn et al., report the characterization of *Aspergillus fumigatus* YakA kinase belonging to the DYRK related family of kinases as an antifungal drug target. First part of the work includes extensive work including annotation and identification of protein kinases (PKs) and generation of the PK knockout mutants to study the contribution of the kinases to fitness. With an objective to identify fungal-specific kinases for drug targeting they identified the YakA kinase (Yak1 ortholog) from a collection of 108 barcoded PK knockout mutants generated. Previous studies have characterized Yak1 kinase orthologs in several other fungi and established their role in regulating

transcription factors under stress conditions, pathogenesis, and in TOR signaling in *Saccharomyces cerevisiae*. Here the authors reveal roles for YakA in stress adaptation (oxidative, pH, temperature, antifungal drug stressors and in the presence or absence of iron).

The novel findings of this study include,

- Identification of a unique role for YakA kinase in regulating septal pore plugging, by phosphorylating the Woronin body tethering Leashin protein (Lah), and Spa10 protein under iron deplete stress conditions.
- They show that the YakA mutant is reduced in its ability to infect lung tissue in a murine model, consistent with reduced ability to penetrate solid agarose.
- They show the efficacy of a DYRK inhibitor, 1-ECBC, in specifically inhibiting YakA function in preventing septal pore blockage and being synergistic with azoles. They further extend this interesting study to structural modeling of YakA and identify the key active site binding pocket for 1-ECBC on YakA and postulate that YakA kinase can potentially be a druggable target for future therapeutic approaches in combination with azoles.

Overall, this work is conceptually and technically sound including the rigorous methods adopted for generating the PK knockouts and CRISPR system used for the essential kinase, fitness assays, and the number of biological replicates taken for the phosphoproteomic analysis experiments to obtain robust dataset of native proteome versus phosphoproteome.

While the authors have presented data towards conceptualizing YakA kinase as a potential fungal specific drug target utilizing specific YakA inhibitor, I have several concerns with the data presented which the authors should consider.

Specific comments:

1. How many mice were taken per group for the murine aspergillosis model? It looks like the *yakA* mutant was mildly avirulent with a mortality of 60% in the neutropenic murine model (Figure 2). Considering the septal plugging as critical for adaptive response to triazole challenge, I think examining the virulence of *yakA* mutant in the background of azole treatment would add value. In addition, the authors have shown that the DYRK inhibitor, 1-ECBC, blocks septal pore recruitment of YakA under iron limitation. As synergism was observed between DYRK inhibitor and azoles, I strongly recommend using 1-ECBC in their murine model experiments.

2. It was not clear as to what were the actual MIC values were for azole for the wild-type and the *yakA* mutant (Figure 2). As the importance of septa for overcoming cell wall stress has been reported I would recommend testing the impact of echinocandins on the *yakA* mutant under iron limitation.

3. While the authors conclude that *A. fumigatus* YakA has been repurposed from other fungal species with unique functions including septal pore plugging, no complementation experiments to show the difference in function between *Aspergillus* YakA versus other species ortholog were performed which would be confirmatory.

4. The section dealing with YakA phosphorylating proteins associated with the septum needs to be elaborated (Figure 4). Phosphorylation does not necessarily mean that the sites are important for function. Mutational studies on the phosphosites identified are required to suggest a direct role for YakA in regulating the candidate septal proteins, Spa10 and LahC.

5. The microscopic images in Figures 5 and 6 are not high resolution and were not clear to me in the PDF file. No scale bars are provided. No quantitative data is provided to claim, "the apparent abundance of YakA at the septa in iron deplete or presence of rapamycin" (Figure 5).

6. Line 602: I do not think that YakA regulates septal plugging of HexA as we still see septal localization of HexA in the absence of YakA (Figure 6). Lines 619-621: Please clarify the statement "Deletion of *yakA*...by the GFP tag". The BF image of LahC:GFP Δ *yakA* does not have good resolution and it does not seem like there is any septa there.

Reviewer #1 (Remarks to the Author):

This study focuses on the identification of *Aspergillus fumigatus* protein kinases that regulate disease-relevant phenotypes in an attempt to uncover novel antifungal drug targets. The disease burden associated with invasive fungal infections is high and treatment outcomes are poor. *A. fumigatus* is the most common cause of deeply invasive mold infections. Currently available antifungal drugs are limited and their continued use is further threatened by recent increases in resistance seen around the globe. For environmental fungi, like *A. fumigatus*, this resistance to the leading class of antifungals (triazoles) is partly driven by agricultural use of azole compounds. The authors report the generation of a barcoded *A. fumigatus* protein kinase deletion library that was subsequently screened in competitive fitness assays by bar-seq. Conditions used for fitness tests either mimicked stresses encountered in the host, or were actually performed in a mouse model. Although several kinases are found to support pathogenic and antifungal fitness, the authors focus on the YakA kinase. Data provided show that the YakA kinase is important for virulence in a mouse model of aspergillosis, characterized by reduced tissue invasion. Data also shows YakA is required for hyphal penetration into high percentage agar; loss of YakA causes reduced phosphorylation of septum-associated proteins; YakA localizes to the septal pore upon low iron stress where it subsequently promotes localization and activation of septal pore-plugging machinery. As the *yakA* deletion is also more susceptible to triazoles, the authors provide evidence that a known inhibitor of a YakA ortholog in yeast blocks septal localization of YakA and synergizes with triazoles *in vitro*. This work is significant for the field, as generation of the mutant library and successful employment in fitness screening could lead to additional finding down the road. This is especially important as novel antifungal targets that might both inhibit virulence but also synergize with current antifungals, are badly needed. Furthermore, protein kinases are readily targeted. The manuscript is fairly well-written, however, there were multiple omissions that have to be addressed. The figures are presented in a clear and logical manner, but the legends largely lack useful detail. Stats seem to be appropriately run. The conclusions are largely supported the data (with exception of issues noted below).

We would like to thank reviewer 1 for their insightful comments, which we have addressed point-by-point below.

1. In general, the Figure Legends could use a bit more detail to keep the reader from continually referencing the MM section for things like stats and replicates.

We have reviewed all figure legends and updated to enhance the technical information provided.

2. Stats are not included on the survival graphs in Figure 2

We have now addressed this oversight and included the stats in figure 2.

3. There is no description of how the *yakA* reconstituted strain was generated.

We have included a section in the materials & methods.

4. It is unclear what is meant by the text in lines 540-543? No data is shown to support this vague statement but the authors use it to argue that YakA function in *A. fumigatus* is divergent from yeast.

We have modified this section of the manuscript so we focus specifically on the domains identified in *S. cerevisiae* and how the *A. fumigatus* protein differs from this. We have also modified the figure to focus specifically on these points.

5. For Figure 6, the difference in localization (focal point at septal pore vs wider localization over septum) is really evident by eye. How many replicates were measured here? The quantitative data only appears to show one septum per condition.

The reviewer is correct the data presented represents a single septal region. While we were not initially intending to perform quantitative evaluations of the plots, we have, thanks to the reviewers prompt, been able to show statistical differences in the distribution of HexA:GFP in the wild-type and *yakA* null mutants. We have similarly analysed the LahC:GFP data.

6. Does loss of yakA actual shift the MIC to triazoles based on EUCAST or CLSI guidelines? The growth assays are fine, but are on solid agar and show minor shift under non-standard conditions. Therefore, they do not correlate well with the checkerboard studies in Figure 7.

Yes the loss of YakA does result in a reproducible and significant change in MIC to Voriconazole. This information has been added to supplemental figure 6

7. There is no description of the conditions under which the checkerboard assay was run.

We have added a description in the materials and methods.

8. Glucose uptake experiments are described in the methods but are found no where in the presented data.

We have removed this section from the materials & methods.

9. What is Supplemental Figure 8A? It is not described.

We have added a description of Supplemental Figure 8A in the legend.

10. Lines 765 and 784: Do the authors mean 1-ECBC?

We have amended these lines.

11. The conclusion that the 1-ECBC/voriconazole synergism is due to blocking YakA localization to the septal pore is not well-supported. 1-ECBC is very likely to have off-target effects in the cell and the study lacks mutants that might help resolve this (YakA overexpression or hyperactive strains, etc.). This is not even offered as a caveat in the Discussion.

We were careful not to state specifically that there is a direct correlative link between 1-ECBC action on YakA and increased voriconazole activity. Establishing this would be exceptionally challenging - and I do not expect that overexpression of yakA would address this as YakA is regulated by the essential protein kinases TorC and PKA – and its not clear to us that the abundance of yakA transcripts are rate limiting. Notably yakA transcription levels do not vary much (based on our unpublished data, and data available at fungiDB) suggesting that transcription abundance is not a natural way of regulating this kinase. To address the reviewers concerns however we have included a section in the discussion to clearly state that while loss of yakA results in azole sensitivity, and ECBC prevents localisation of yakA to the septal pore, care must be taken when inferring direct causation from this data as ECBC is likely to have off target effects.

12. In this line of thought (#11), do aseptate mutants show the same hypersusceptibility to triazoles? These were previously published to not display triazole phenotypes (by CLSI MIC) in a previous study using a protein kinase disruption library (referenced by the authors). However, the authors are working in a different strain background. If these are in the authors collection and showed the same phenotype(s), it would be very supportive that it is actually the blocking of septal plugging by 1-ECBC treatment or loss of yakA that generates the triazole susceptibility phenotype.

To address the reviews point we have assessed the susceptibility of a *hexA* null mutant, which is unable to block its septal pore. This mutant phenocopies the YakA mutant with respect to azole sensitivity. This information is shown in Supplemental Figure 6 and text has been included in the results section.

Reviewer #2 (Remarks to the Author):

After deletion analysis of protein kinase genes in *Aspergillus fumigatus* a potential novel antifungal drug target was identified. Using competitive fitness profiling the potential target is shown to enhance sensitivity to azoles and has defects in fitness within models of infection. The target, yakA, locates and plays roles at septal pores and is involved is septal pore plugging and the ability of cells to

penetrate solid media and grow in mice lung tissues. A yak1 inhibitory drug is shown to also affect septal plugging and synergize with azoles to inhibit *A. fumigatus* growth. Proteomic analysis identifies some potential targets of the kinase including septal pore proteins consistent with yakA potentially regulating septal pore functions. Overall, there are some important findings presented in this study and the data are overall clear and conclusions reached well-reasoned, with a few exceptions as detail further below. The manuscript could be improved by making the overall presented work easier to understand and with the many minor defects listed below considered.

More important issues:

L 49. Further data would need to be collected to conclude definitively that YakA regulates blocking of the septal pore upon exposure to stress via phosphorylation of the Woronin body tethering protein Lah. At the moment only a correlation is shown. Alternatively, the conclusions should be softened to not claim causation.

We have amended this statement.

L 451. I could not find Supplementary Table 1. Assume what is meant is Supplemental Data 1, Library sheet. It would be more reader friendly to give better headings for all the Supplemental Data. For example, there are 5 sheets in Suppl data 1, none of which are labeled within the text. Similarly, there are 22 sheets in Supplemental Data 3 none of which are specifically mentioned in the text, all are referred to as Supplemental Data 3. The supplemental data should also be further reviewed as color designations are not explained and column headings are missing or difficult to understand.

We have amended references of Supplementary Table 1 to Supplemental Data 1. The Supplemental Data files have been amended to give more clarity.

L 451. Was there a reason not to use heterokaryon rescue to confirm that lack of recovery after gene deletion was actually caused because the target genes were essential, (PMID: 19883779 for example) rather than some other more mundane technical reason? There are methods to demonstrate heterokaryons are formed when an essential gene is deleted carrying both nuclei with the WT allele and nuclei with the null allele and also define the phenotype caused by the null mutation.

The reviewer makes a valid point with respect to heterokaryon rescue. Indeed, as part of the efforts to generate the null mutants, we isolated, where possible, 3 putative transformants for the vast majority of the 'notionally' essential genes identified and confirmed that upon sub-culture they were unable to grow on selective media. We have added this data to the manuscript and to supplemental table 1 (under heading heterokaryon rescue) and included reference to this in the methods section. We have removed direct reference to gene essentiality as we believe that heterokaryon rescue is inconclusive – and positive confirmation via use of an inducible promoter system provides more robust evidence.

Suppl. Fig. 5B. The effect of agarose level on Δ yakA is not fully explained by saying it is exacerbated under iron limitation. There seems to be a graded, yet reversible, effect which is intriguing but not discussed. Does the fact that inhibition appears to be equally high at the lowest and highest levels of agarose fit with the model of turgor pressure issues?

Many thanks to the reviewer for highlighting an omission in our description of this figure. We hypothesise that the apparent recovery of the yakA null phenotype when agarose of 1-4% is used is due to trace element (likely iron) contamination in the agarose. Despite assessing agarose from many different suppliers we have been unable to source material that is completely iron free. We have added comment to this effect in the figure legend.

L 626. Not sure it is possible to conclude that HexA depends upon YakA for location at the septal pore, it is there without yakA (Fig. 6). Further, no mention is made of the distribution pattern across the septum showing a central peak and two shoulder peaks. What are the shoulder peaks? This pattern is not discernable from the images presented and should be shown. Is the quantitation from a single imaging section or from a projection, same question for all the images shown. The peak likely represents the actual septal pore region (6B) and would suggest that there is increased HapA-GFP around this region without YakA, not a decrease, as concluded.

We understand the reviewers point about our statement in line 626. To clarify, when not localised to septal pores, HexA is found in peroxisomes (see <https://doi.org/10.1016/j.ijmm.2012.11.005>). Peroxisomal targeting of HexA is mediated by a peroxisomal targeting sequence (PTS1) at its C-terminus. The act of fusing the GFP tag at the C-terminus blocks the PTS1 sequence and hence HexA no longer localises to the peroxisomes, rather it localises across the septa (when not at the septal pore). Importantly the HexA-GFP fusion does not localise at the septal pore when the C-terminal domain of Lah is truncated (lah Δ C see figure 6 B and C in <https://onlinelibrary.wiley.com/doi/full/10.1111/mmi.12316>). Critically our experiments support a role for yakA in mediating this as localisation of the HexA-GFP fusion in the yakA null phenocopies what is seen in lah Δ C. To support our statement further, and to address the other issues raised by the reviewer, we have performed a quantitative analysis across three septa to show a statistically significant difference in localisation of HexA, away from the septal pore, occurs in the absence of yakA. Furthermore we have included additional images to more conclusively show that hexA is no longer localised to the septal pore in the yakA null.

Because HexA-GFP localises to septal pore regions in low iron (6A) with or without yakA, how can it be concluded that YakA activity is required for micronutrient stress-initiated localization of Lah and HexA at the septal pore and subsequent septal plugging?

As highlighted above, our data demonstrates that HexA only localises to the septal pore where there is a functional YakA. This is consistent with what is shown in the lah Δ C (figure 6 B and C in <https://onlinelibrary.wiley.com/doi/full/10.1111/mmi.12316>)

L 693. Several things in this paragraph are unclear. What is the difference in the leukopenic murine model of infection vs neutropenic mouse infection model vs murine model?

We have clarified all references to our models so that we are more precise. We now refer to mouse models rather than murine (mouse and rat) models, similarly we have used the term leukopenic rather than neutropenic (although the terms are regularly used interchangeably).

L 697, what data is equally compelling?

We have attempted to clarify our statement.

L 703, It is unclear what ...distribution of fitness values for replicates in the mouse model was bimodal... means? How can this be explained by induced hyphal fusion? Further details should be given for this argument.

We have attempted to clarify this.

Figure 7B. In the bottom panels, after shift from low iron to low iron + ECBC, it appears that the cell lyses at 15'. Is this a common event and is it dependent on the drug or due to the media shift. Would the effects be different if media exchange was done without added drug? This would seem to be an important control.

We have added Supplemental Movie 1 and 2, which show that this is precipitate floating into frame. Hopefully this clears up that it is not lysis.

L 646. "Upon iron limitation 1-ECBC blocks YakA recruitment to the septal pore, which in turn blocks LahC recruitment (Figure 7B)." No data is apparent regarding the effects of 1-ECBC on LahC. Deletion of Yak1 and stopping its location to septa using a drug are two separate things.

We have amended this statement.

Other issues:

L 35. Is *A. fumigatus* a pathogen or an opportunistic one?

A. fumigatus is a pathogen. The term opportunistic pathogen has, rightly, come under some criticism see <https://bmcbiol.biomedcentral.com/articles/10.1186/1741-7007-10-6>

Fig 1 A. Grey circles not obviously grey, too dark.

We have changed this in the legend to clarify. The statement of colour is not required to interpret this figure.

L 41. ...120 genetically barcoded null mutants.... L 102. ...108 genetically barcoded protein kinase null mutants... This discrepancy in the number of mutants should be explained for clarity.

We have amended our text to refer to 108.

L 404. Unclear what is meant by, ..now defunct annotations of the *A. fumigatus* genome..

We have amended this statement to highlight that we are referring to previous versions.

Was there any correlation with the essential protein kinase genes previously defined in *A. nidulans* to those predicted in *A. fumigatus*?

Yes the majority of the protein kinases for which we were unable to isolate null mutants were also essential or conditionally essential in *A. nidulans*. This data has been added to the results section and to supplementary data table 1.

L 456. Do the authors have an explanation for this discrepancy in their ability to obtain some null allele strains? Might need some further clarification/explanation.

We have added a potential clarification for the discrepancy between the studies. The study by de souza et al in *A. fumigatus* used a gene disruption approach rather than a gene replacement approach. It is likely that the mutants generated in the previous study retained some functionality for the target kinase.

Fig. 2A. 2A is somewhat impenetrable. The gene designations are difficult to read, gene names are missing and designations are not searchable. It is unclear how the competitive fitness growth experiments were completed. Were all strains inoculated together at one time?

We have increased the text size so that it is more legible. The figure will be provided in high resolution so that the zoom function can be used. We will also provide the figure in a format that permits search function (i.e. pdf) in the final submitted version. We have included additional information for the bar-seq experiment in the methods section and the main body of the text.

L 1086. Wile should be while

Change made.

Suppl Fig 2. What is FusC?

We have added this to the legend.

L 523. Yak1 has also been described and deleted in *A. nidulans*.

We have added reference to this work in our manuscript however we find little evidence that it has been described in *A. nidulans* in any meaningful way.

Suppl. Fig. 3. Pom1/2 is not mentioned in the manuscript main text.

We have removed this.

L 587. Should it be Suppl Data 3?

Correct and changed.

Fig. 5B 6B. Is the scale correct (μm)? If so, these would be unusually wide cells that happened to be 100 μm wide. Also, for the graphs of the GFP signals at septa, what is the n and was any statistical analysis completed?

The scale should be distance in %, we have amended these. We choose % to normalise for difference in hyphal width from hyphae to hyphae. We have added more replicates and details in the legend. We completed a statistical analysis.

Size bars should be added to all cell image figures. At the moment, some are missing and some way too small to see, assuming the tiny bars in some images are in fact size bars?

We have added appropriate scale bars.

L 632. Is adaptation the correct word here?

We have changed this to susceptibility.

L642. What does figure refer to in (ΔG : -7.7 kcal/mol; figure)?

We have made this change.

L 1078. Is shown the correct word here?

Yes

L 1129. GF-HexA needs correcting.

We have changed this.

Supplemental Figure 8, nothing indicated regarding 8A.

We have made this change.

L 646. Text regarding Supplemental Figure 8A does not match what is shown in the actual figure. Some quantitation of data shown in Fig. 7B could be given.

We have made this change.

L 647. ... which is turn... should be ..which in turn..

This sentence has been removed.

L 649. "... unblocking of the septal pore is a YakA independent process..." Not sure how this argument can be made when no unblocking of septal pores is shown.

We have amended this statement as we are presenting an hypothesis here and not a conclusion.

FICI should be defined.

We have added a definition.

L. 667. Seems be does not fit in this sentence.

We are unable to find these words in this sentence.

L 677. Mixing of nuclei between cells should also be mentioned.

We have added this.

L 785. Should the drug not be 1-ECBC rather than 1-ABC?

Correct and has been amended.

Reviewer #3 (Remarks to the Author):

In this manuscript van Rhijn et al., report the characterization of *Aspergillus fumigatus* YakA kinase belonging to the DYRK related family of kinases as an antifungal drug target. First part of the work includes extensive work including annotation and identification of protein kinases (PKs) and generation of the PK knockout mutants to study the contribution of the kinases to fitness. With an objective to identify fungal-specific kinases for drug targeting they identified the YakA kinase (Yak1 ortholog) from a collection of 108 barcoded PK knockout mutants generated. Previous studies have characterized Yak1 kinase orthologs in several other fungi and established their role in regulating transcription factors under stress conditions, pathogenesis, and in TOR signaling in *Saccharomyces cerevisiae*. Here the authors reveal roles for YakA in stress adaptation (oxidative, pH, temperature, antifungal drug stressors and in the presence or absence of iron).

The novel findings of this study include,

- Identification of a unique role for YakA kinase in regulating septal pore plugging, by phosphorylating the Woronin body tethering Leashin protein (Lah), and Spa10 protein under iron deplete stress conditions.
- They show that the YakA mutant is reduced in its ability to infect lung tissue in a murine model, consistent with reduced ability to penetrate solid agarose.
- They show the efficacy of a DYRK inhibitor, 1-ECBC, in specifically inhibiting YakA function in preventing septal pore blockage and being synergistic with azoles. They further extend this interesting study to structural modeling of YakA and identify the key active site binding pocket for 1-ECBC on YakA and postulate that YakA kinase can potentially be a druggable target for future therapeutic approaches in combination with azoles.

Overall, this work is conceptually and technically sound including the rigorous methods adopted for generating the PK knockouts and CRISPR system used for the essential kinase, fitness assays, and the number of biological replicates taken for the phosphoproteomic analysis experiments to obtain robust dataset of native proteome versus phosphoproteome.

While the authors have presented data towards conceptualizing YakA kinase as a potential fungal specific drug target utilizing specific YakA inhibitor, I have several concerns with the data presented which the authors should consider.

Specific comments:

1. How many mice were taken per group for the murine aspergillosis model? It looks like the yakA mutant was mildly avirulent with a mortality of 60% in the neutropenic murine model (Figure 2).

Considering the septal plugging as critical for adaptive response to triazole challenge, I think examining the virulence of yakA mutant in the background of azole treatment would add value. In addition, the authors have shown that the DYRK inhibitor, 1-ECBC, blocks septal pore recruitment of YakA under iron limitation. As synergism was observed between DYRK inhibitor and azoles, I strongly recommend using 1-ECBC in their murine model experiments.

We are planning to perform further murine model experimentation to explore the properties of 1-ECBC. However, this work is would be exceptionally extensive, requiring toxicity studies with mammalian cells, safety evaluation in mice, optimisation of dosing of ECBC and drug combinations, co-toxicity studies, PK and tissue distribution (and development of analytical methods to detect ECBC in murine tissues) etc. This would require a significant investment of resource and time (>1 year) and may reveal that 1-ECBC is not a good drug candidate. While interesting and worth pursuing, this is well beyond the scope of this manuscript.

2. It was not clear as to what were the actual MIC values were for azole for the wild-type and the yakA mutant (Figure 2). As the importance of septa for overcoming cell wall stress has been reported I would recommend testing the impact of echinocandins on the yakA mutant under iron limitation.

We have added data showing the impact of loss of yakA on echinocandin susceptibility and further data on the consequences of loss of *hexA* on azole susceptibility in supp figure 6.

3. While the authors conclude that *A. fumigatus* YakA has been repurposed from other fungal species with unique functions including septal pore plugging, no complementation experiments to show the difference in function between *Aspergillus* YakA versus other species ortholog were performed which would be confirmatory.

To clarify we are not suggesting that yakA has been horizontally transferred from other species, rather it has evolved a function different from that seen in other fungi. Since many yeasts do not have septa (as they are unicellular and don't have hyphae), and many other fungi do not have Woronin bodies, it is safe to say that yakA must have been repurposed.

4. The section dealing with YakA phosphorylating proteins associated with the septum needs to be elaborated (Figure 4). Phosphorylation does not necessarily mean that the sites are important for function. Mutational studies on the phosphosites identified are required to suggest a direct role for YakA in regulating the candidate septal proteins, Spa10 and LahC.

We agree with the reviewer – we use the phosphoproteomics data to provide indicative information about the function of yakA (which we then follow up with additional experimentation) but do not directly state that the sites are directly involved in the process of septal plugging. We are keen to identify which of the phosphorylation sites are important for the regulation of septal plugging however there are several proteins of interest with each having many different phosphosites which maybe functionally redundant. The task requested is monumental. Inferring functionality of these sites and performing these experiments is currently therefore beyond the scope of this paper.

5. The microscopic images in Figures 5 and 6 are not high resolution and were not clear to me in the PDF file. No scale bars are provided. No quantitative data is provided to claim, “the apparent abundance of YakA at the septa in iron deplete or presence of rapamycin” (Figure 5).

We have added scale bars and quantification of the fluorescent signal in conditions mentioned.

6. Line 602: I do not think that YakA regulates septal plugging of HexA as we still see septal localization of HexA in the absence of YakA (Figure 6).

See the answer to the previous reviewers question.

Lines 619-621: Please clarify the statement “Deletion of yakA...by the GFP tag”. The BF image of LahC:GFP Δ yakA does not have good resolution and it does not seem like there is any septa there.

We have replaced the image with a better quality image showing a septum.